# Acetylation dynamics and stoichiometry in *Saccharomyces cerevisiae*

Brian T. Weinert[1], Vytautas Iesmantavicius[1], Tarek Moustafa[2], Christian Schölz[1], Sebastian A. Wagner[1], Christoph Magnes[3], Rudolf Zechner[2] & Chunaram Choudhary[1,*]

## Abstract

**Lysine acetylation is a frequently occurring posttranslational modification; however, little is known about the origin and regulation of most sites. Here we used quantitative mass spectrometry to analyze acetylation dynamics and stoichiometry in *Saccharomyces cerevisiae*. We found that acetylation accumulated in growth-arrested cells in a manner that depended on acetyl-CoA generation in distinct subcellular compartments. Mitochondrial acetylation levels correlated with acetyl-CoA concentration *in vivo* and acetyl-CoA acetylated lysine residues nonenzymatically *in vitro*. We developed a method to estimate acetylation stoichiometry and found that the vast majority of mitochondrial and cytoplasmic acetylation had a very low stoichiometry. However, mitochondrial acetylation occurred at a significantly higher basal level than cytoplasmic acetylation, consistent with the distinct acetylation dynamics and higher acetyl-CoA concentration in mitochondria. High stoichiometry acetylation occurred mostly on histones, proteins present in histone acetyltransferase and deacetylase complexes, and on transcription factors. These data show that a majority of acetylation occurs at very low levels in exponentially growing yeast and is uniformly affected by exposure to acetyl-CoA.**

**Keywords** acetylation; mass spectrometry; mitochondria; proteomics; stoichiometry
**Subject Categories** Post-translational Modifications, Proteolysis & Proteomics; Genome-Scale & Integrative Biology
**Mol Syst Biol. (2014) 10: 716**

## Introduction

Lysine acetylation is an evolutionarily conserved, reversible posttranslational modification (PTM) that is known to regulate protein function by site-specific acetylation (catalyzed by acetyltransferases) and deacetylation (catalyzed by deacetylases). Acetyltransferases use acetyl-coenzyme A (acetyl-CoA) as a cofactor donating an acetyl group to the ε-amino group of lysine. Acetylation is a well-known regulatory PTM in the context of nuclear signaling, in particular for regulating gene expression via modification of histones. The role of acetylation in regulating nuclear processes is consistent with the nuclear localization of most acetyltransferases and proteins with acetyllysine-binding bromodomains. A prominent role for non-nuclear acetylation was suggested by numerous proteomic studies showing that lysine acetylation occurs at thousands of sites throughout eukaryotic cells (Kim *et al*, 2006; Choudhary *et al*, 2009; Zhao *et al*, 2010; Weinert *et al*, 2011; Chen *et al*, 2012; Henriksen *et al*, 2012; Lundby *et al*, 2012; Hebert *et al*, 2013).

Notably, acetylation has been shown to occur frequently on mitochondrial proteins and a large number of sites are regulated by the sirtuin-class deacetylase-3 (SIRT3; Sol *et al*, 2012; Hebert *et al*, 2013; Rardin *et al*, 2013). These observations, and a number of supporting studies (Newman *et al*, 2012), have established acetylation as an important regulator of mitochondrial metabolism. However, the mechanisms by which acetylation occurs in this organelle are mostly unknown. Several recent reviews speculated that nonenzymatic acetylation by acetyl-CoA may be widespread in mitochondria (Guan & Xiong, 2010; Newman *et al*, 2012). Acetylation could regulate metabolism through a nonenzymatic mechanism, in which case determining the stoichiometry of acetylation is of critical importance for gauging the effects of acetylation on protein function.

In this study we used quantitative mass spectrometry (MS) to measure changes in lysine acetylation abundance in *Saccharomyces cerevisiae* (budding yeast). We found that growth-arrest, combined with ongoing metabolism, resulted in the accumulation of acetylation in a manner that depended on acetyl-CoA generation in distinct subcellular compartments. Acetylation dynamics in mitochondria correlated with acetyl-CoA levels in this compartment and we found that acetyl-CoA nonenzymatically acetylated protein *in vitro*. We developed a novel method to estimate acetylation stoichiometry and discovered that the vast majority of acetylation occurred at a very low level. These data provide the first global analysis of acetylation stoichiometry and indicate that metabolism regulates acetylation levels in distinct subcellular compartments through acetyl-CoA generation.

1 The NNF Center for Protein Research, Faculty of Health Sciences, University of Copenhagen, Copenhagen, Denmark
2 Institute of Molecular Biosciences, University of Graz, Graz, Austria
3 HEALTH – Institute for Biomedicine and Health Sciences, Joanneum Research, Graz, Austria
*Corresponding author. Tel: +45 35 32 50 20; Fax: +45 35 32 50 01; E-mail: chuna.choudhary@cpr.ku.dk

# Results

## Experimental strategy for quantitative analysis of proteome and PTM dynamics

We used stable isotope labeling with amino acids in cell culture (SILAC; Ong *et al*, 2002) to quantify differences in protein, acetylation, and phosphorylation abundance by MS. Proteins from whole cell lysates were digested to peptides and acetylated peptides enriched using a polyclonal anti-acetyllysine antibody as previously described (Kim *et al*, 2006; Weinert *et al*, 2013a). Peptide fractions were analyzed by reversed-phase liquid chromatography coupled to high resolution liquid chromatography-tandem mass spectrometry (LC-MS/MS) and raw MS data were computationally processed using MaxQuant (Cox *et al*, 2011). The quantitative MS experiments performed in this study are summarized in Supplementary Table S1, and the raw data files have been deposited to the ProteomeXchange (Vizcaino *et al*, 2013), with the identifier PXD000507.

## Acetylation dynamics in growth-arrested yeast cells

In previous work we showed that acetylation accumulates in growth-arrested bacteria (*E. coli*) cells due to prolonged exposure to, and a higher concentration of, acetyl-phosphate (Weinert *et al*, 2013a). In order to test whether growth-arrest affects PTMs in a eukaryotic system we compared acetylation, phosphorylation, and protein levels between exponentially growing and stationary phase *S. cerevisiae* cells. We quantified more than 2600 acetylation sites (Supplementary Table S2), 3300 proteins (Supplementary Table S3), and 6000 phosphorylation sites (Supplementary Table S4) with a high correlation between biological replicates (Supplementary Figs S1A–C). Strikingly, stationary phase cells showed increased ( > 2-fold) acetylation at a majority (~70%) of quantified acetylation sites (median threefold increased; Fig 1). In contrast, protein and phosphorylation abundance, while affected in stationary phase cells, was not globally increased (Fig 1), indicating that stationary phase did not cause the accumulation of proteins or PTMs generally. Furthermore, Gene Ontology (GO) enrichment analysis of protein and phosphorylation site changes revealed both up-regulation and down-regulation of specific processes in stationary phase cells (Supplementary Figs S1D and E), suggesting that such changes occurred in a regulated manner. Using subcellular localization data from GFP-tagged proteins (Huh *et al*, 2003), we found that acetylation increased to the greatest degree on mitochondrial proteins (median 5.7-fold), which was significantly higher than the median increases seen for cytoplasmic (2.6-fold) and nuclear (1.5-fold) proteins (Fig 1). Increased acetylation of mitochondrial proteins was nearly comprehensive (93% of sites were > 2-fold elevated; Fig 1 and Supplementary Table S2).

During glycolysis, glucose is converted to pyruvate, which enters mitochondria and is further converted to acetyl-CoA by the pyruvate dehydrogenase complex (PDC), pyruvate can also be converted to acetyl-CoA in the cytoplasm via an acetate intermediate (Fig 2A). In order to control the transition to stationary phase we growth-arrested exponentially growing cultures by transferring the cells into media containing different carbon sources, and lacking an essential amino acid (lysine), for ~24 h. Notably, mitochondrial acetylation

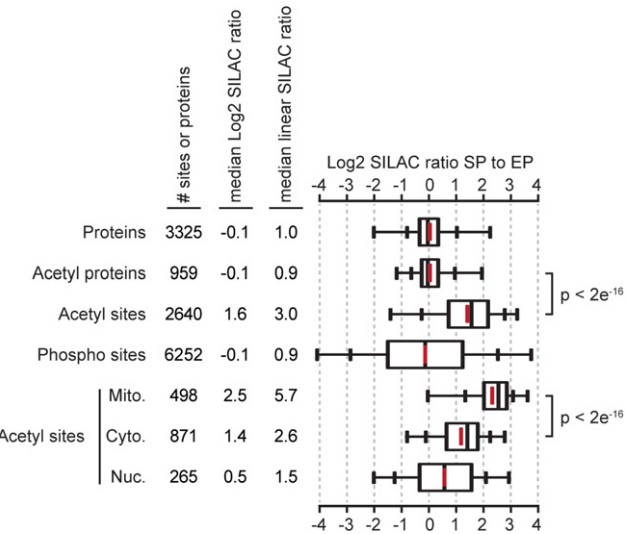

**Figure 1. Acetylation is globally increased in stationary phase yeast cells.**
Box plots showing the distributions of SILAC ratios comparing stationary phase (SP) to exponential phase (EP) yeast for all quantified proteins (proteins), acetylated proteins (acetyl proteins), acetylation sites corrected for protein abundance changes (acetyl sites), and phosphorylation sites (phospho sites). The box portion of the plot indicates the middle 50% of the distribution, inner hatch marks denote 9–91%, and whisker ends 2–98%, outliers are not shown. Acetylation sites occurring on proteins localized exclusively to mitochondria (Mito.), the cytoplasm (Cyto.), or the nucleus (Nuc.) are shown. Statistical significance was calculated using a Wilcoxon test, data is from two biological replicates.

was unaffected when cells were growth-arrested in the presence of 2-deoxy-D-glucose (2DG; a glucose analog that cannot be metabolized beyond the first step of glycolysis), indicating that increased mitochondrial acetylation depended on glycolysis (Fig 2B, Supplementary Fig S2A, and Supplementary Table S5). However, nuclear acetylation was globally reduced in the presence of 2DG (median 3.6-fold), suggesting that nuclear deacetylases remain active under these conditions.

In order to examine the dependence of mitochondrial protein acetylation on glycolysis, we quantified acetylation dynamics in *pda1Δ* cells that are unable to convert pyruvate to acetyl-CoA (Wenzel *et al*, 1992), or in *cit1Δ* cells that are unable to convert acetyl-CoA to citrate (Kispal *et al*, 1988; Fig 2A). Mitochondrial acetylation was drastically reduced in exponentially growing *pda1Δ* cells (median SILAC ratio *pda1Δ*/wild-type = 0.14) and was slightly increased in exponentially growing *cit1Δ* cells (Fig 2C, Supplementary Fig S2B, and Supplementary Table S6). Loss of Pda1 completely suppressed the increased acetylation of mitochondrial proteins in growth-arrested cells while loss of Cit1 further increased the acetylation of mitochondrial proteins (an additional 1.8-fold) in growth-arrested cells (Fig 2D, Supplementary Fig S2C, and Supplementary Table S7). Loss of Cit1 blocks the entry of acetyl-CoA into the citric acid cycle, thus, further increased acetylation in growth-arrested *cit1Δ* cells is likely due to greater accumulation of acetyl-CoA in the mitochondria of these cells. We quantified half as many acetylation sites on mitochondrial proteins in *pda1Δ* cells compared to *cit1Δ* cells, while the frequency of quantified sites on cytoplasmic

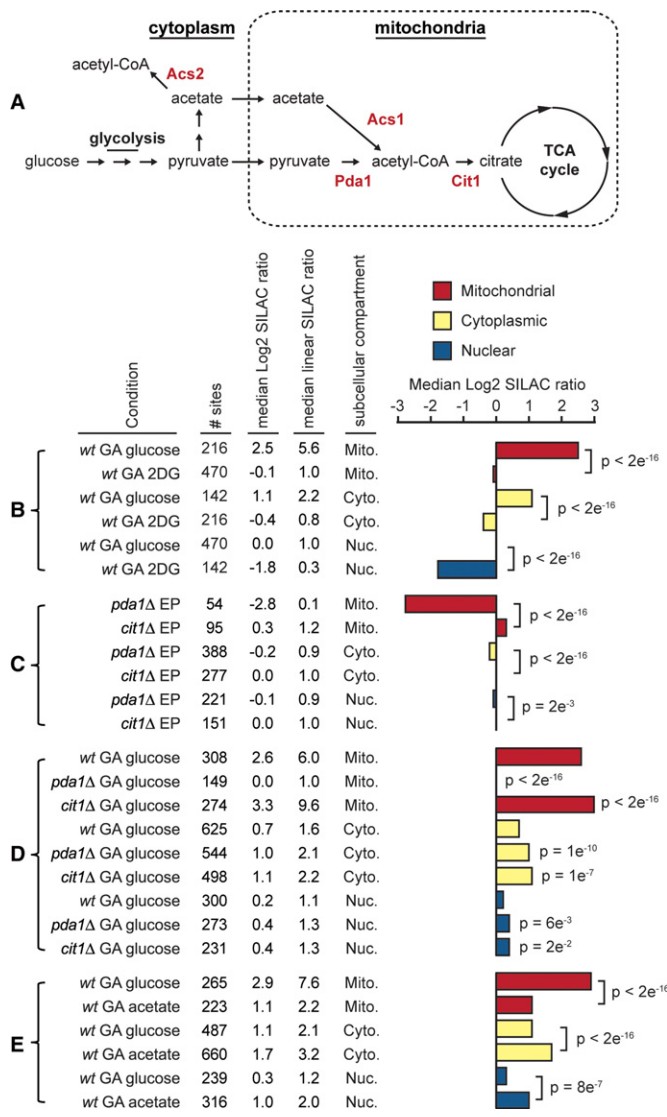

**Figure 2.    Quantitative analysis of acetylation dynamics in yeast.**

A    Model showing the formation of acetyl-CoA from glucose and acetate, key enzymes are shown in red type.

B–E    The figure shows the conditions analyzed in each experiment, the cell type (wild-type (*wt*) or indicated mutant strains), the growth state [exponential phase (EP) or growth-arrested (GA)], the number of acetylation sites analyzed (# sites), the median Log2 and linear SILAC ratios comparing the indicated condition to wild-type EP cells, and the subcellular localization of the analyzed acetylation sites on proteins localized to mitochondria (Mito.), the cytoplasm (Cyto.), or the nucleus (Nuc.). Cells were growth-arrested by transferring an exponential phase culture into media lacking lysine and containing the indicated carbon sources, glucose, acetate, or 2-deoxy-ᴅ-glucose (2DG). The bar chart shows the median Log2 SILAC ratios comparing the indicated condition to wild-type EP cells, statistical significance was calculated by Wilcoxon test. Increased acetylation requires glycolysis (B). Mitochondrial acetylation in exponentially growing cells requires Pda1 (C). Increased mitochondrial acetylation in growth-arrested cells is suppressed by loss of Pda1 and enhanced by loss of Cit1 (D). Data from two biological replicates is shown, significant differences are relative to wild-type cells. Acetate promotes cytoplasmic and nuclear acetylation (E). Data is from two biological replicates.

and nuclear proteins was not altered (Figs 2C and D), indicating that acetylation of many sites in *pda1Δ* cells had diminished to undetectable levels in mitochondria specifically.

We next compared growth-arrested yeast in the presence of glucose or acetate to test whether acetate would alter the acetylation dynamics in growth-arrested cells. Consistent with conversion of acetate to acetyl-CoA by acetyl-CoA synthetase 2 (Acs2; Fig 2A; Takahashi *et al*, 2006) in the cytoplasm, acetate preferentially promoted the acetylation of cytoplasmic and nuclear proteins (Fig 2E, Supplementary Fig S2D, and Supplementary Table S8).

## Acetylation levels in mitochondria correlate with acetyl-CoA concentration

We showed that mitochondrial acetylation was suppressed by deletion of *pda1*, suggesting that most mitochondrial acetyl-CoA is generated in a Pda1-dependent manner in the growth conditions used in our study. We compared acetyl-CoA levels between wild-type, *pda1Δ*, and *cit1Δ* cells (Fig 3A). Acetyl-CoA was reduced by approximately 1/3 in *pda1Δ* cells, indicating that the mitochondrial acetyl-CoA pool constitutes at least 1/3 of the total acetyl-CoA pool. However, mitochondria occupy ~1–2% of the total cellular volume in yeast (Uchida *et al*, 2011), indicating that acetyl-CoA levels are substantially (~20–30-fold) higher in mitochondria compared to the cytoplasm and nucleus. We hypothesized that increased acetylation in growth-arrested cells may occur due to higher concentrations of acetyl-CoA or due to prolonged exposure of proteins to acetyl-CoA. We found that acetyl-CoA levels were reduced in growth-arrested cells (Fig 3A), suggesting that acetylation occurs due to prolonged exposure to acetyl-CoA. However, this difference may be due to reduced recovery of acetyl-CoA from growth-arrested cells, which have a substantially increased cell wall and are known to be refractory to cell lysis. We consistently recovered less protein from growth-arrested cells (data not shown), suggesting that these cells were more difficult to lyse, or that cell size and/or protein content was reduced under these conditions. Such differences in cell physiology may explain the lower amount of acetyl-CoA per $OD_{600}$ of growth-arrested cells as determined by our assay. Regardless, acetyl-CoA levels were elevated in growth-arrested *cit1Δ* cells, consistent with our observation of increased acetylation in these cells. If we assume that the reduction of acetyl-CoA in *pda1Δ* cells indicates the mitochondrial acetyl-CoA pool in these cells, then mitochondrial acetyl-CoA in *cit1Δ* cells was increased ~1.9-fold and ~1.7-fold after 8 and 16 h growth-arrest, respectively (Fig 3A), mirroring the 1.8-fold increase in mitochondrial acetylation in growth-arrested *cit1Δ* cells (Fig 2D).

The physiological concentration of acetyl-CoA in yeast is not well-studied. One study estimated acetyl-CoA levels (3–30 μM) in nutrient-starved yeast undergoing metabolic cycles (Cai *et al*, 2011), conditions that contrast with the excess of glucose and high growth rates of yeast grown on synthetic complete (SC) media in our study. We estimated a similar cellular concentration of ~30 μM acetyl-CoA in exponentially growing cells. However, this estimate assumes complete recovery of acetyl-CoA, and is therefore likely to underestimate the actual cellular concentration. We showed that mitochondrial acetyl-CoA was ~20–30-fold higher than non-mitochondrial acetyl-CoA, suggesting a concentration of acetyl-CoA in mitochondria that approaches the millimolar range (~0.5–1 mM

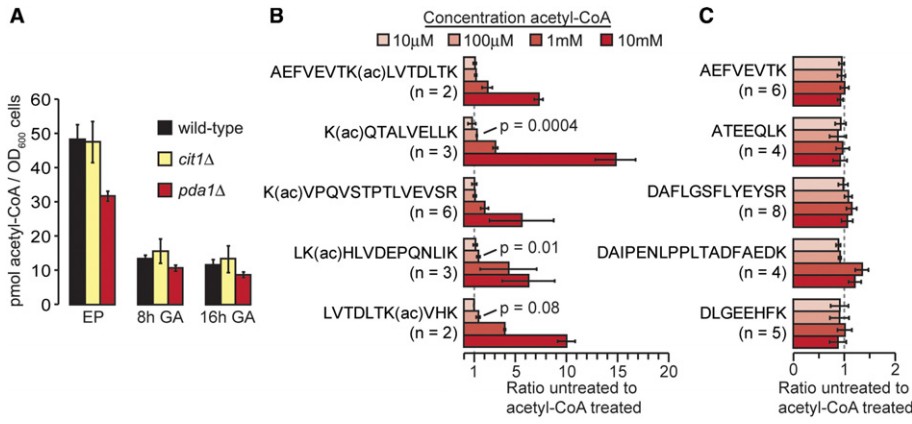

**Figure 3.  Acetyl-CoA concentration in cells and nonenzymatic acetylation by acetyl-CoA.**

A    Acetyl-CoA concentration was determined in the indicated cell types during exponential phase (EP) growth or after the indicated time of growth-arrest (GA) in the presence of glucose. Data are from two independent biological replicates.

B    The bar graph shows the abundance of the indicated, acetylated (ac) peptides relative to an untreated control sample. Error bars indicate standard deviation of the indicated number (n) of independently quantified peptides. The significance (P) of increased acetylation at 100 μM acetyl-CoA was calculated by two-tailed t-test assuming equal variance.

C    The column graph shows the relative abundance of non-acetylated peptides. Error bars indicate standard deviation of the indicated number (n) of independently quantified peptides.

based on our estimates). This estimate is consistent with previous work showing that acetyl-CoA can reach millimolar levels in rat liver mitochondria (Garland et al, 1965).

### Acetyl-CoA nonenzymatically acetylates protein

A study in 1970 showed that acetyl-CoA could nonenzymatically acetylate lysine rich histone fractions and synthetic polylysine at a concentration of ~0.5 mM, with increased kinetics at higher pH, and an activation energy of 7.5 kcal, suggesting that the reaction could occur under physiological conditions (Paik et al, 1970). We sought to independently verify nonenzymatic acetylation by acetyl-CoA using mass spectrometry and a quantitative strategy based on isobaric mass tags (Supplementary Fig S3). Bovine serum albumin (BSA) was treated with increasing concentrations of acetyl-CoA at physiological pH, modestly, but significantly, increased acetylation was seen on several peptides at a concentration of 100 μM acetyl-CoA, while 1 and 10 mM acetyl-CoA caused increased acetylation on all peptides analyzed (Fig 3B). In contrast, the abundance of non-acetylated peptides was unaffected by acetyl-CoA (Fig 3C). This acetylation is likely to be nonenzymatic since we treated a purified extracellular serum protein (BSA) and acetylation did not occur until acetyl-CoA concentration was well in excess of the known range of binding affinities of acetyltransferases for acetyl-CoA (Albaugh et al, 2011a).

### Acetylation stoichiometry in yeast

The striking increase (median 9.6-fold) in mitochondrial acetylation in growth-arrested cit1Δ cells indicated that mitochondrial acetylation occurred at a low-level (maximum median stoichiometry of ~10% in exponentially growing cells). One method used to analyze PTM stoichiometry on a global scale is to compare the relative abundances of posttranslationally modified and unmodified corresponding peptides (CPs; Olsen et al, 2010; Wu et al, 2011). If the abundance of a modified peptide is substantially altered then the abundance of the CP should be affected in an inverse manner. However, CP abundance will only be measurably affected if PTM stoichiometry is relatively high. Since our results suggested a low stoichiometry of modification, we devised a strategy to assay acetylation stoichiometry based on partial chemical acetylation of lysine residues. Partial chemical acetylation will cause the greatest relative increases in acetylation on sites with the lowest initial stoichiometry, while sites with higher initial stoichiometry will be partially or not at all affected. In order to find conditions that resulted in partial chemical acetylation, we chemically acetylated bovine serum albumin (BSA) with acetyl-phosphate (AcP) or acetic anhydride. We found that AcP caused partial chemical acetylation that did not measurably affect the abundance of CPs, while acetic anhydride caused comprehensive acetylation and resulted in dramatically reduced abundance of CPs (Supplementary Fig S4 and supplemental text).

In order to examine acetylation site stoichiometry in S. cerevisiae, we used a SILAC-based quantitative approach to quantify chemically acetylated and untreated peptides in the same MS experiment (Fig 4). An untreated control sample was prepared in order to identify acetylation sites that occurred in the absence of AcP-treatment and to determine the quantitative variability in our experiments. Treatment with AcP resulted in substantially increased (>2-fold) acetylation at 74% (10 mM AcP) and 88% (100 mM AcP) of acetylation sites (Fig 5A and Supplementary Table S9). In contrast, <1% of acetylation sites were substantially increased in the untreated (0 mM AcP) lysate, reflecting the quantitative accuracy of our assays (Fig 5A). An independent experimental replicate was performed with untreated (0 mM) and 100 mM AcP-treated lysate (Supplementary Table S9). Acetylation changes were similar in experimental replicates, as shown by a Spearman's correlation

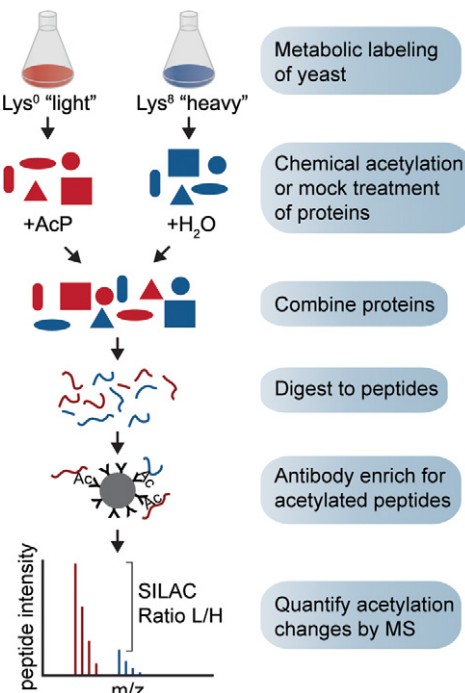

**Figure 4.  Method used to assay acetyl-phosphate sensitivity in yeast lysate.**
Yeast proteins were metabolically labeled with unlabeled "light" lysine or with "heavy" isotope labeled lysine. Protein lysates were treated with acetyl-phosphate (AcP) or were mock-treated by addition of $H_2O$. Equal amounts of protein were mixed and digested to peptides with trypsin protease. Acetylated (Ac) peptides were immune-affinity enriched using agarose-coupled anti-acetyllysine polyclonal antibody and analyzed by MS. Increased acetylation from AcP treatment causes an increase in the relative abundance of the light labeled peptide, enabling quantification of acetylation changes induced by AcP.

coefficient of 0.89 (Supplementary Fig S5). We identified 9415 acetylation sites in 100 mM AcP-treated lysates, 1765 sites had SILAC ratios, and 916 of these sites were previously identified (Supplementary Table S2), or were found in untreated control samples, indicating that these sites are naturally occurring acetylation sites in yeast.

Using only the 916 naturally occurring sites with SILAC ratios, we found that 65% of the sites were more than 10-fold increased in acetylation (Fig 5B), indicating maximum initial stoichiometry that was < 10% at these sites. However, acetylation stoichiometry at these positions is likely to be lower than 10% (which assumes comprehensive acetylation) because our analysis of AcP-treated BSA (Supplementary Fig S4), showed that 100 mM AcP caused partial chemical acetylation without a reduction in the abundance of CPs. CP abundance was similarly unaffected in 100 mM AcP-treated yeast lysate (Supplementary Fig S6), indicating that AcP-treatment caused a low-level of partial chemical acetylation.

In order to better characterize the degree of partial chemical acetylation by 100 mM AcP, we used an absolute quantification (AQUA) method (Gerber *et al*, 2003), to quantify acetylation levels. Acetylated and unmodified peptide intensities were compared to heavy-labeled peptide standards, allowing us to determine acetylation stoichiometry directly. We analyzed eight acetylation sites with varying sensitivity to 100 mM AcP on two proteins, 3-phosphoglycerate kinase (Pgk1)

and fatty acid synthetase (Fas2; Table 1). Surprisingly, we were unable to detect acetylated peptides from purified Pgk1, and found just one acetylated peptide from purified Fas2, even though we detected unmodified peptides covering 89 and 72% of the Pgk1 and Fas2 protein sequences, respectively. In order to detect acetylation on these proteins we treated cell lysates with 100 mM AcP and purified Pgk1 and Fas2 for AQUA analysis. Comparison to heavy-labeled peptide standards indicated that acetylation stoichiometry, after AcP-treatment, was < 1% at all eight sites, with a median degree of chemical acetylation that was just 0.07% (Table 1, Supplementary Figs S7A–D). AcP-sensitivity was well-correlated with acetylation stoichiometry as determined by the AQUA method (Spearman's correlation of −0.92, Fig 5C), confirming our prediction that low stoichiometry sites would be most sensitive to partial chemical acetylation and providing independent validation of our method.

We estimated acetylation stoichiometry based on the conservative assumption that chemical acetylation from 100 mM AcP was < 1% at all sites. A site with 10-fold increased acetylation after AcP-treatment was estimated to have a stoichiometry that is < 0.1% while a site with 20-fold increased acetylation was estimated to have a stoichiometry that is < 0.05%. Using this approach we estimated acetylation stoichiometry based on the observed SILAC L/H ratios after treatment with 100 mM AcP and found that 25% of sites were < 0.02% acetylated, 50% of sites were < 0.05% acetylated, 66% of sites were < 0.1% acetylated, and a remarkable 86% of sites were < 1% acetylated (Fig 5D). These estimates contrast with previously determined phosphorylation site stoichiometries in yeast, where 89% of sites were estimated to have stoichiometries that were >1% (Wu *et al*, 2011; Fig 5E), indicating that a much greater fraction of acetylation occurs with a low stoichiometry. It was not possible to accurately estimate the stoichiometry of sites that were AcP-insensitive (SILAC ratio L/H < 2) as the relative changes in acetylation were of a similar magnitude to the variability of these measurements. Thus, AcP-insensitive sites were estimated to have acetylation stoichiometry that is >1%.

In order identify independent parameters that could confirm our stoichiometry estimates we compared acetylated peptide intensity and abundance-corrected acetylated peptide intensity to AcP-sensitivity (Supplementary Fig S8). We found that correcting acetylated peptide intensity with protein abundances determined by an intensity-based absolute quantification (iBAQ) method (Schwanhausser *et al*, 2011), could best distinguish AcP-insensitive sites (SILAC ratio L/H < 2) from AcP-sensitive sites (SILAC ratio L/H > 2; Fig 5F and Supplementary Fig S8) and had the best overall correlation (ρ = −0.57) with AcP sensitivity (Supplementary Fig S8). These data independently confirmed our stoichiometry estimates based on AcP-sensitivity, as this parameter was calculated using peptide intensities and protein abundances determined in untreated control samples.

We estimated acetylation stoichiometry for the 916 sites with SILAC ratios; however, an additional 1540 sites without SILAC ratios were independently identified as naturally occurring sites (identified in untreated control samples or in Supplementary Table S2). SILAC ratios were estimated for these sites by calculating the increased intensity of AcP-treated "light" peptides relative to an empirically determined detection limit for naturally occurring "heavy" peptides (see Materials and Methods). Using this method we calculated minimum ratios of increased acetylation at 1516

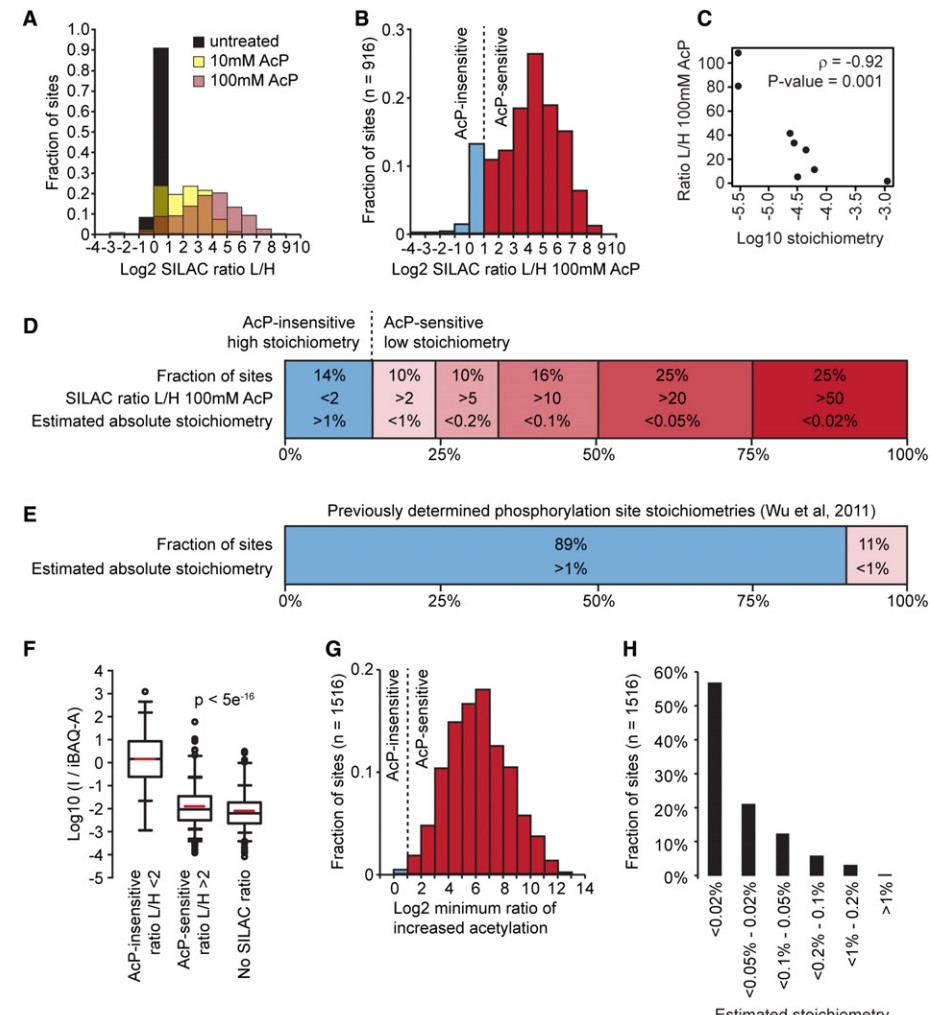

**Figure 5.  Most acetylation sites are modified with a low stoichiometry.**

A    The majority of yeast acetylation sites are highly sensitive to partial chemical acetylation by AcP. The histogram shows the distribution of SILAC L/H ratios for the indicated samples.

B    AcP caused substantially increased acetylation at a majority of sites. The histogram shows acetylation changes induced by 100 mM AcP in two experimental replicates, only sites that were independently identified in cells without AcP treatment are shown.

C    Acetylation stoichiometry is inversely proportional to AcP-sensitivity. The scatterplot shows the relationship between AcP-sensitivity (SILAC ratio L/H 100 mM AcP) and acetylation stoichiometry (Log10 stoichiometry) as determined by AQUA analysis (Table 1). The Spearmans correlation ($\rho$) and the significance by two-tailed test (*P*-value) are shown.

D    Most acetylation occurs with a low stoichiometry. Absolute acetylation site stoichiometries were estimated based on relative abundance changes (SILAC ratio L/H) after treatment with 100 mM AcP and using an estimate that AcP treatment caused < 1% chemical acetylation.

E    For comparison, previously determined phosphorylation site stoichiometries are shown (Wu *et al*, 2011).

F    iBAQ-based abundance corrected acetylated peptide intensity (I/iBAQ-A) is proportional to AcP sensitivity. The box plots show the distributions of I/iBAQ-A values for the indicated classes of acetylation sites. AcP-insensitive sites had a significantly (*p*) higher distribution of I/iBAQ-A values compared to either AcP-sensitive sites or sites without SILAC ratios. Significance was calculated by Wilcoxen test.

G    Sites without SILAC ratios are highly sensitive to AcP. The minimum ratio of increased acetylation was determined by calculating the increased intensity of AcP-treated "light" peptides relative to an empirically determined detection limit for "heavy" SILAC peptides (see Materials and Methods).

H    Absolute acetylation stoichiometry of sites without SILAC ratios was estimated to be very low. Stoichiometry was estimated by the same method used to estimate stoichiometry of sites with SILAC ratios in (D) using the minimum ratios of increased acetylation shown in (G).

sites and found that AcP caused a median 70-fold increase in acetylation (Fig 5G). Assuming that 100 mM AcP caused < 1% chemical acetylation, the substantially increased acetylation at these sites indicated a very low stoichiometry of acetylation (Fig 5H) and this finding was independently verified by examining the

abundance-corrected peptide intensity (I/iBAQ-A) distributions for these sites (Fig 5F). By including these data we estimated acetylation stoichiometry at a total of 2432 sites in exponentially growing yeast and found that 95% of all acetylation occurs at an estimated stoichiometry that is < 1%.

**Table 1.** Absolute quantification (AQUA) of acetylated peptides from Pgk1 and Fas2. Acetylation stoichiometry was determined by AQUA for Pgk1 and Fas2 after treatment with 100 mM AcP (see Supplementary Fig S6). Initial stoichiometry was calculated by dividing the stoichiometry in 100 mM AcP by the Ratio L/H 100 mM AcP and the degree of chemical acetylation is the difference between the initial stoichiometry and the stoichiometry in 100 mM AcP

| Protein | Peptide sequence | Stoichiometry 100 mM AcP (%) | Ratio L/H 100 mM AcP | Initial stoichiometry (%) | Degree chemical acetylation (%) |
|---|---|---|---|---|---|
| Pgk1 | AAGFLLEK(ac)ELK | 0.023 | 80.8 | 0.0003 | 0.02 |
| Pgk1 | AGAEIVPK(ac)LMEK | 0.095 | 41.5 | 0.0023 | 0.09 |
| Pgk1 | FAAGTK(ac)ALLDEVVK | 0.119 | 27.7 | 0.0043 | 0.11 |
| Fas2 | QVLDVDPVYKDVA(ac)PTGPK | 0.032 | 108.3 | 0.0003 | 0.03 |
| Fas2 | LIEPELFNGYNPE(ac)K | 0.092 | 33.5 | 0.0027 | 0.09 |
| Fas2 | GATLYIP(ac)ALR | 0.068 | 11.3 | 0.0060 | 0.06 |
| Fas2 | SEGNPVIGVFQ(ac)FLTGHPK | 0.016 | 5.3 | 0.0031 | 0.01 |
| Fas2 | AND(ac)NESATINEMMK | 0.181 | 1.7 | 0.1055 | 0.08 |

## Functional analysis of proteins with high stoichiometry acetylation sites

In order to characterize proteins with AcP-insensitive, high stoichiometry sites we used GO term enrichment analysis. Proteins with AcP-insensitive sites were significantly more frequently associated with nuclear processes (Fig 6A). More than half of all AcP-insensitive sites occurred on histones, proteins present in histone deacetylase (HDAC) or histone acetyltransferase (HAT) complexes, and on transcription factors. We compared the effect of AcP-treatment on proteins present in different subcellular compartments (Fig 6B). Consistent with GO term analysis, 77% of the high-stoichiometry, AcP-insensitive sites occurred on nuclear proteins. In contrast, 97% of mitochondrial and 94% of cytoplasmic acetylation sites were AcP-sensitive, indicating that the vast majority of acetylation sites in these subcellular compartments were acetylated at a low stoichiometry. Notably, mitochondrial acetylation sites were significantly ($p = 5.8e^{-11}$, Wilcoxon test) less affected (median 14-fold increased) by AcP than cytoplasmic sites (median 30-fold increased; Fig 6B), suggesting a higher basal level of acetylation in mitochondria. This finding was independently confirmed by a significantly higher distribution of acetylated peptide I/iBAQ-A for mitochondrial proteins than cytoplasmic proteins (Fig 6C). In contrast, non-acetylated peptide I/iBAQ-A was similarly distributed between mitochondrial and cytoplasmic proteins (Fig 6C). It is also notable that the distribution of AcP-sensitivity for nuclear proteins was bi-modal, indicating distinct populations of high and low stoichiometry acetylation sites in the nucleus. The subcellular localization of high-stoichiometry, AcP-insensitive sites was significantly different to low-stoichiometry, AcP-sensitive sites and low-stoichiometry sites without SILAC ratios in all three compartments (Fig 6D).

MS analysis is biased towards abundant proteins, which are more readily detected in the MS. Nearly every abundant protein identified by MS was also acetylated (Fig 6E), and the frequency of detected acetylation sites was proportional to protein abundance (Supplementary Fig S9A), indicating that protein abundance is a limiting factor in the identification of many sites. The technical bias to detect acetylation on abundant proteins was less pronounced for mitochondrial proteins compared with cytoplasmic proteins, consistent with our estimation of higher acetylation levels in mitochondria

(Fig 6E). Similarly, nuclear proteins were least biased to detect acetylation on abundant proteins, as this subcellular compartment contained the most high stoichiometry acetylation sites (Fig 6E). Thus, the abundance of proteins with observed acetylation sites is consistent with our stoichiometry estimates.

Acetylated sites with high stoichiometry are likely to be important for protein function. Many previously known regulatory acetylation sites were AcP-insensitive, including Smc3 lysines 112 and 113 (Zhang *et al*, 2008), Sas2 (MYST homolog) lysine 168 (Yuan *et al*, 2012), Yng2 lysine 170 (Lin *et al*, 2008), RTT109 lysine 290 (Albaugh *et al*, 2011b), SNF2 lysines 1494 and 1498 (Kim *et al*, 2010), histone H2AZ (Htz1) lysines 4, 9, 11, and 15 (Babiarz *et al*, 2006; Millar *et al*, 2006; Lin *et al*, 2008), histone H3 (Hht1) lysines 19, 24, and 57 (Zhang *et al*, 1998; Suka *et al*, 2001; Hyland *et al*, 2005), histone H4 (Hhf1) lysines 6, 9, 13, and 17 (Suka *et al*, 2001) and histone H2B (Htb2) lysines 16 and 17. These 20 sites constitute 18% of the 111 AcP-insensitive sites that we identified, indicating that AcP-insensitivity is a good predictor of functionally important acetylation sites. The enrichment of functionally characterized sites within the group of AcP-insensitive sites is highly significant ($p = 1e^{-15}$, Fisher exact test). Furthermore, known functional sites had significantly higher I/iBAQ-A values, indicating that this parameter is also useful in distinguishing functionally-relevant, high-stoichiometry sites (Supplementary Fig S9B and Supplementary Table S9).

## Discussion

While acetylation has been identified on thousands of proteins in diverse subcellular compartments, the mechanisms of non-nuclear acetylation and its regulation are not well understood. We found that acetylation levels were dynamically affected by growth-arrest and the generation of acetyl-CoA in distinct subcellular compartments. Acetyl-CoA was previously shown to exist in distinct mitochondrial and non-mitochondrial pools in yeast (Takahashi *et al*, 2006). We showed that mitochondrial acetylation occurs within mitochondria and is uniformly affected by growth-arrest and the generation of acetyl-CoA by the PDC. We found that acetyl-CoA levels correlated with acetylation changes in mitochondria and we

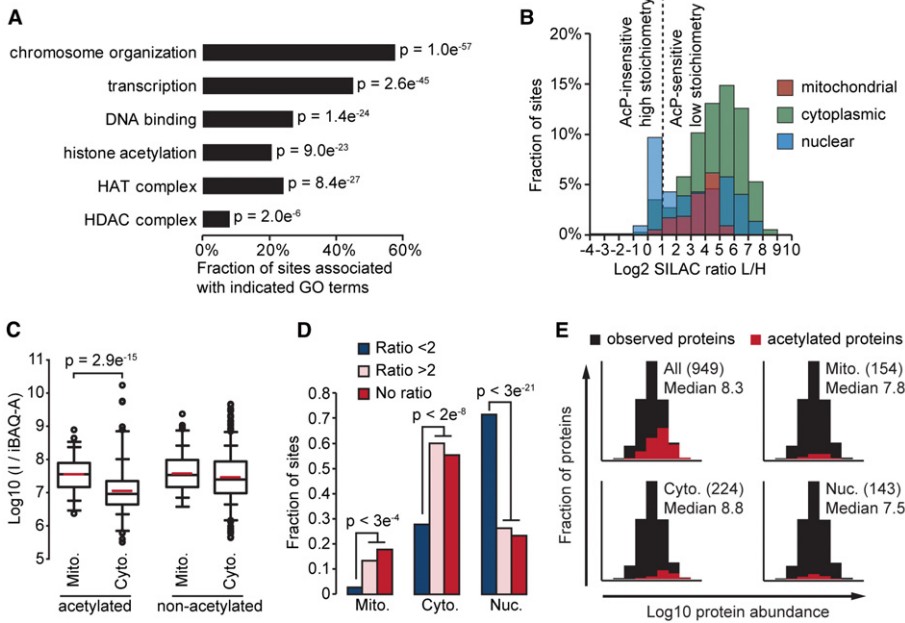

**Figure 6.    Functional analysis of high stoichiometry acetylation sites.**

A    AcP-insensitive acetylation sites occur on proteins associated with nuclear processes. Gene Ontology (GO) term enrichment was performed by comparing proteins with AcP-insensitive acetylation sites (ratio L/H < 2) to all acetylated proteins. The bar graph shows the percentage of AcP-insensitive sites occurring on proteins associated with the indicated GO terms. *P*-values (*P*) indicate the statistical significance of GO term enrichment by Fishers exact test.

B    AcP-insensitive acetylation sites occur on nuclear proteins. The histogram shows the distribution of SILAC L/H ratios occurring on proteins localized to the indicated subcellular compartments.

C    Acetylated peptides from mitochondrial proteins have a significantly higher median I/iBAQ-A value compared to acetylated peptides from cytoplasmic proteins. The box plots show the I/iBAQ-A distributions for the indicated classes of peptides occurring on mitochondrial (Mito.) or cytoplasmic (Cyto.) proteins. Signficance (*p*) was determined by Wilcoxon test.

D    AcP-insensitive sites (SILAC Ratio < 2) have a significantly different subcellular distribution. Sites without SILAC ratios (No ratio) have a similar subcellular distribution to AcP-sensitive sites (SILAC Ratio > 2). The bar graph shows the fraction of sites localized to the indicated subcellular compartments; mitochondria (Mito.), cytoplasm (Cyto.), or nucleus (Nuc.). Significance (*P*) was calculated by Fisher exact test.

E    Detection of acetylation sites is biased to occur on abundant proteins and this bias is more pronounced for sites with the lowest estimated stoichiometries. The histograms show the distributions of iBAQ protein abundances for observed proteins (*n* = 3,104). The distributions of the indicated classes of acetylated proteins occurring exclusively in the indicated subcellular compartments is shown in red. The numbers of acetylated proteins are shown in parenthesis and the median Log10 iBAQ abundance for these acetylated proteins is shown.

confirmed that acetyl-CoA can nonenzymatically acetylate proteins. Our analysis of acetylation stoichiometry showed that the vast majority of acetylation occurs at very low levels, an important observation with widespread implications for understanding non-nuclear acetylation.

A few recent reviews speculated that acetylation may occur nonenzymatically by exposure to acetyl-CoA, particularly in mitochondria where a protein acetyltransferase activity has not been definitively identified (Guan & Xiong, 2010; Newman *et al*, 2012). Acetyl-CoA has been previously shown to nonenzymatically acetylate proteins (Paik *et al*, 1970) and the elevated pH in the mitochondrial matrix would favor this reaction (Paik *et al*, 1970; Llopis *et al*, 1998; Abad *et al*, 2004; Wagner & Payne, 2013). In addition, recent work showed that lysines can be nonenzymatically modified by other glycolytic intermediates, such as 1,3-bisphosphoglycerate (Moellering & Cravatt, 2013), and succinyl-CoA (Wagner & Payne, 2013; Weinert *et al*, 2013b). We found that nearly all acetylation occurred at a low-level and was uniformly regulated by manipulations that affected the generation of acetyl-CoA in distinct subcellu-

lar compartments. Mutations that blocked acetyl-CoA generation or its downstream conversion to citrate had opposing effects on mitochondrial acetylation in growth-arrested yeast cells, linking acetyl-CoA levels to mitochondrial acetylation levels. We furthermore showed that acetyl-CoA levels correlated with acetylation abundance in mitochondria. Growth-arrest in the presence of glucose or acetate affected acetylation levels in a manner that was consistent with separate mitochondrial and non-mitochondrial acetyl-CoA pools and generation. Mitochondrial proteins had a significantly higher basal level of acetylation than cytoplasmic proteins, consistent with our results indicating a higher concentration of acetyl-CoA in this organelle. Acetylation is widespread in yeast (Henriksen *et al*, 2012), yet the number of known acetyltransferases in yeast is small (~15; Cherry *et al*, 2012), particularly compared to the number of kinases catalyzing protein phosphorylation (~130; Manning *et al*, 2002; Cherry *et al*, 2012). Moreover, most acetyltransferases have known nuclear functions, consistent with our observation that the majority of high stoichiometry acetylation occurred in the nucleus. Thus, the global acetylation dynamics that

we observed in growth-arrested yeast, their dependence on distinct acetyl-CoA pools and generation, a preponderance of low stoichiometry acetylation, and the restriction of high stoichiometry acetylation to the nucleus, are highly consistent with a nonenzymatic mechanism of acetylation. These observations contrast with the canonical model of protein regulation by site-specific PTMs and indicate that acetylation may act in a global manner to regulate proteins, or conversely, that most acetylation occurs as a low-level protein lesion that accumulates under specific conditions. However, widespread low-level acetylation could occur due to a promiscuous acetyltransferase activity that is sensitive to changes in acetyl-CoA levels.

To better understand our estimates of acetylation stoichiometry, it is important to note that individual lysines may vary in the degree of chemical acetylation by AcP. Without knowing the exact degree of chemical acetylation at each position it is not possible to make a precise estimate of acetylation stoichiometry. In order to account for this variability we based our estimates on the conservative assumption that the degree of partial chemical acetylation by AcP was <1%. Since we determined that 100 mM AcP caused a median chemical acetylation of just 0.07% (between 0.01 and 0.11%), we are likely to overestimate the stoichiometry of acetylation. Thus, stoichiometry estimates are presented as less than (<) values, to designate an estimated stoichiometry that is less than the indicated amount. In addition, some lysines may be inaccessible or otherwise unreactive, resulting in an incorrect estimate of high stoichiometry acetylation. Regardless of these limitations, the results obtained using our approach indicated that it was able to distinguish acetylation stoichiometry at the site level. Absolute quantification indicated that acetylation stoichiometry was inversely proportional to AcP-sensitivity (Table 1, Fig 5C). The highly significant bias to identify high stoichiometry sites on nuclear proteins indicated that the identification of such sites was non-random. Similarly, sites we predicted to have high stoichiometry were significantly more likely to have known functional roles. Our method further distinguished between low stoichiometry acetylation in the cytoplasm and mitochondria, indicating that the difference in acetylation levels between these two subcellular compartments was greater than the inherent variability of lysine reactivity with AcP. Thus, the resolution of our assay was not limited by the variability in lysine reactivity.

Phosphorylation is known to regulate proteins in a site-selective manner and is one of the most extensively studied and widespread PTMs in eukaryotic cells. The number of detected phosphorylation sites increases with organism complexity, reflecting its role in regulating diverse cellular processes. In contrast, acetylation occurs as frequently in bacteria as in human cells (Weinert *et al*, 2013a). Similar to bacteria, acetylation is overrepresented in mitochondria compared with phosphorylation (Weinert *et al*, 2011). Thus the frequency of acetylation is uncoupled from organism complexity, suggesting that nonenzymatic acetylation may play a large role in the generation of these sites. We showed that basal acetylation levels are elevated in mitochondria, likely due to a higher concentration of acetyl-CoA in this organelle. These observations suggest that the endosymbiotic evolution of mitochondria as the metabolic centers of eukaryotic cells may have limited widespread acetylation outside of mitochondria by compartmentalizing acetyl-CoA generated during metabolism, thereby enabling the evolution of acetylation signaling in the nucleus.

## Materials and Methods

### Yeast growth and lysate preparation

Wild-type *Saccharomyces cerevisiae* cells (BY4742, MATalpha his3Δ1 leu2Δ0 lys2Δ0 ura3Δ0; ThermoFisher Scientific, Slangerup, Denmark), *pda1Δ* (BY4742, pda1::KanMX; ThermoFisher Scientific), and *cit1Δ* (BY4742, cit1::KanMX; Open Biosystems) were cultured in synthetic complete media (US Biological, Salem, MA, USA) supplemented with $^{12}C_6^{14}N_2$-lysine (SILAC "light") or $^{13}C_6^{15}N_2$-lysine (SILAC "heavy"). Cells were harvested at the indicated time points, washed once with sterile $H_2O$, and resuspended in lysis buffer (50 mM Tris, pH7.5, 150 mM NaCl, 1 mM EDTA, 1x mini complete protease inhibitor cocktail (Roche, Basel, Switzerland), 5 mM sodium fluoride, 1 mM sodium orthovanadate, 5 mM beta-glycerophosphate, 10 mM nicotinamide, and 5 μM tricostatin A) at ~50 $OD_{600}$ cells/ml lysis buffer. The cell suspension was frozen drop-wise in liquid nitrogen and ground in a liquid nitrogen chilled steel container by the Retsch MM 400 Ball Mill (Retsch, Haan, Germany) for 5 min at 25 Hz. The lysate was thawed, NP-40 and sodium deoxycholate were added to a final concentration of 1 and 0.1%, respectively, and clarified by centrifugation. The lysate supernatent was precipitated with four volumes −20°C acetone. The acetone precipitate was dissolved in urea solution (6 M urea, 2 M thio-urea, 10 mM Hepes pH8.0) and protein concentration determined by Quick-Start Bradford assay (Bio-Rad, Copenhagen, Denmark).

### Chemical acetylation of BSA and yeast lysate

For treatment of BSA with acetyl-CoA, BSA was prepared in PBS (pH 7.2) at 10 mg/ml and mixed with 1/10 volume acetyl-CoA (Sigma, Copenhagen, Denmark) prepared fresh in $H_2O$. Reactions were incubated at 30°C for 4 h and stopped by addition of nine volumes −20°C acetone. Acetone precipitated protein was resuspended in 200 μl of 100 mM triethyl ammonium bicarbonate (TEAB) and digested to peptides by addition of 1/100 (w/w) trypsin protease (Sigma) and incubation at 37°C for 16 h. Tryptic peptides were labeled with TMT sixplex isobaric mass tags (ThermoFisher Scientific) according to the manufacturer's instructions. Chemical acetylation with acetic anhydride was performed as described (Guan *et al*, 2010). Briefly, 20 ml of 1 mg/ml BSA (Sigma) was prepared in 0.1 M $Na_2CO_3$. 100 μl acetic anhydride (Sigma) was added slowly over 10 min in a glass beaker with constant stirring, followed by 400 μl pyridine (Sigma) over 30 min, also with constant stirring. The reaction was then left to incubate at room temperature for 4 h and quenched by addition of 400 μl 1 M Tris-base (Sigma). Organic solvents were removed by five rounds of centrifuge filtration in an Amicon Ultra 15 (Millipore, Billerica, MA, USA) with a 10 kD molecular weight cutoff. For treatment with AcP, BSA was prepared in yeast lysis buffer (as above, with NP-40 and deoxycholate added) at a concentration of 20 mg/ml. Yeast lysate was similarly prepared at a concentration of 20 mg/ml. BSA or yeast lysate was then mixed with 1/10 volume of $H_2O$ or AcP [freshly prepared in $H_2O$, potassium lithium salt (Sigma)] at either 100 mM or 1 M concentration for a final concentration of 10 or 100 mM AcP, respectively. Reactions were incubated at 37°C for 90 min. BSA was mixed with 4x nuPAGE SDS-polyacrylamide loading buffer (Life Technologies,

Naerum, Denmark) and separated by SDS-polyacrylamide gel electrophoresis using NuPAGE gels (Life Technologies). In-gel tryptic digestion and peptide isolation were performed essentially as described (Shevchenko *et al*, 1996). For yeast lysate, the reactions were stopped by addition of 5–10 volumes −20°C acetone and incubated at −20°C for 1–2 h. Acetone precipitated protein was resuspended in 8 M urea solution (6 M urea, 2 M thio-urea, 10 mM Hepes pH8.0) and the concentration determined by Quick Start Bradford assay (BioRad). Equal amounts of protein were mixed, diluted to 2 M urea by addition of 3 volume 50 mM Hepes, pH8.5, proteolyzed by addition of 1:200 (w/w) trypsin protease (Sigma) for 16–20 h at room temperature. Peptides were purified by C18 Sep-Pack classic cartridge (Waters, Waltham, MA, USA). Peptides were acidified by addition of trifluoroacetic acid (TFA) to 1% and applied to a Sep-Pack column that was pre-equilibrated with 5 ml acetonitrile and twice with 5 ml 0.1% TFA. The column was washed twice with 5 ml 0.1% TFA and once with 5 ml $H_2O$. Peptides were eluted with 3 ml 50% acetonitrile in $H_2O$, 100 μl of 10x IP buffer (500 mM MOPS, pH 7.2, 500 mM NaCl) was added, and the acetonitrile was removed by vacuum centrifugation to a final volume of ~1 ml. Peptide concentration was determined by absorbance at 280 nm using a nanodrop (ThermoFisher Scientific) spectrophotometer (~6 mg/ml). An aliquot of 200 μg peptides was set aside for subsequent proteome analysis and the remaining peptides were used for acetyllysine enrichment (described below).

## Peptide preparation, acetyllysine enrichment, and peptide fractionation

In-solution protein digestion, peptide purification, and acetyllysine peptide enrichment were performed essentially as described (Kim *et al*, 2006; Weinert *et al*, 2013a). Immuno-enriched peptides were eluted from anti-acetyllysine antibody resin using acidified $H_2O$ [0.2% trifluoroacetic acid (TFA)]. Peptide eluates were loaded directly onto a strong cation exchange (SCX) microtip column prepared as described (Rappsilber *et al*, 2007; Wisniewski *et al*, 2009). Peptide eluates were briefly evaporated to remove acetonitrile and then loaded onto C18 stage-tips as described (Rappsilber *et al*, 2007). Proteome measurements were made by fractionation of total peptides by the SCX microcolumn method and analyzed by mass spectrometry.

## Quantitative mass spectrometric analysis

We used stable isotope labeling with amino acids in cell culture (SILAC; Ong *et al*, 2002), to measure changes in protein, lysine acetylation, and phosphorylation abundance. Peptide fractions were analyzed by online nanoflow LC-MS/MS using a Proxeon easy nLC system (ThermoFisher Scientific) connected to an LTQ Orbitrap Velos (ThermoFisher Scientific) or Q-Exactive (ThermoFisher Scientific) mass spectrometer. The LTQ Orbitrap Velos instrument was operated under Xcalibur 2.1 (ThermoFisher Scientific) with the LTQ Orbitrap Tune Plus Developers Kit version 2.6.0.1042 software in the data dependent mode to automatically switch between MS and MS/MS acquisition as described (Weinert *et al*, 2011). The Q-Exactive was operated using Xcalibur 2.2 (ThermoFisher Scientific) in the data dependent mode to automatically switch between MS and MS/MS acquisition as described (Michalski *et al*, 2011; Kelstrup *et al*,

2012). All quantitative MS experiments performed in this study are summarized in Supplementary Table S1. The mass spectrometry proteomics data have been deposited to the ProteomeXchange Consortium (http://proteomecentral.proteomexchange.org) via the PRIDE partner repository (Vizcaino *et al*, 2013) with the dataset identifier PXD000507.

## Peptide identification and computational analysis

Raw data files were processed using MaxQuant software (developer version 1.2.7.1) as described (http://www.maxquant.org/; Cox *et al*, 2011). Parent ion (MS) and fragment (MS2) spectra were searched against the Saccharomyces Genome Database (SGD) genome release r63, January 5, 2010. The search was performed using the integrated Andromeda search engine and both forward and reversed (decoy) versions of the databases (Cox *et al*, 2011). Peptides were additionally filtered for a minimum posterior error probability (PEP) score of 0.01, resulting in data sets with estimated false discovery rates that were less than the standard 1% used in most proteomic studies (Supplementary Table S1). Quantification of TMT mass tags may be inaccurate if additional peptides are co-isolated with the targeted peptide ion (Ting *et al*, 2011). In order to minimize this effect we restricted the quantification of TMT mass tags to MS/MS scans in which the targeted parent ion constituted a minimum of 90% of the total ion current. Mass recalibration was performed using high confidence identifications based on a initial "first search" using a 20 part per million (ppm) mass tolerance for parent ion masses and 20 ppm (HCD) or 0.5 Dalton (CID) for fragment ions. Spectra were subsequently searched with a mass tolerance of 6 ppm for parent ions and 20 ppm (HCD) or 0.5 Dalton (CID) for fragment ions, with strict trypsin specificity, and allowing up to two missed cleavage sites. Cysteine carbamidomethylation was searched as a fixed modification, whereas *N*-acetyl protein and oxidized methionine were searched as variable modifications. Where appropriate, acetyllysine was added as a variable modification.

## Acetyl-CoA assay

Frozen yeast pellets (~100 $OD_{600}$ nm) were deproteinized in 400 μl 1 N perchloric acid (PCA) containing 13C2 acetyl-CoA as an internal standard (ISTD), mixed with 200 μl 0.4 mm acid washed glass-beads and vortexed for 2 min at 4°C. Homogenates were centrifuged for 10 min, 10 000 g at 4°C. Supernatant (200 μl) was neutralized by adding repeated aliquots (10 μl) of 3 M KHCO3 under constant vortexing, until bubble evolution ceases (90 μl in total). KClO4 was pelleted by centrifugation for 5 min, 10 000 g at 4°C. Acetyl-CoA from the supernatant was measured as described previously, using ISTD for calibration (Magnes *et al*, 2008). Results were normalized to OD 600 nm measured after extraction.

## AQUA analysis

Yeast strains expressing GFP-tagged Pgk1 and Fas2 at endogenous levels from the chromosomal locus (Huh *et al*, 2003) were obtained from Life Technologies. Protein extracts were prepared exactly as above when treating whole cell lysate with AcP. After treatment with AcP (or mock-treatment) the GFP-tagged proteins were enriched using GFP-trap affinity resin (ChromoTek, Martinsried,

Germany) and the proteins eluted by boiling in 2x tris glycine sample buffer (Life Technologies). Proteins were resolved on pre-cast tris glycine gels (Life Technologies) and peptides recovered by standard in-gel digestion (Shevchenko *et al*, 1996). Heavy-labeled AQUA peptide standards (AQUA quant pro; ThermoFisher Scientific) were mixed with Pgk1 and Fas2 peptides at the indicated concentrations and analyzed by MS. Untreated protein was analyzed in four independent experiments and AcP-treated protein in two independent experiments, all with comparable results.

### Data analysis

Gene Ontology (GO) association and enrichment analysis was performed using the Database for Annotation, Visualization and Integrated Discovery (DAVID) v6.7 (da Huang *et al*, 2009). Statistical tests were performed using R (http://www.r-project.org/index.html). Box plots were generated using Sparklines for Excel (http://sparklines-excel.blogspot.com/). The detection limit for naturally occurring acetylated peptides used to calculate minimum increased acetylation in Fig 5G was determined by ranking the observed "heavy" labeled peptides by peptide intensity and calculating the median intensity of the bottom 10%. This intensity value was an order of magnitude higher than the least intense "heavy" peptide observed in our experiments, thus the estimates of increased acetylation based on this detection limit are conservative.

**Supplementary information** for this article is available online: http://msb.embopress.org

## Acknowledgements

We thank the members of the department of proteomics at CPR for their helpful discussions. We thank the PRIDE team for helping make our data accessible to everybody. This work is supported by the European Commission's 7th Framework Program grants Proteomics Research Infrastructure Maximizing knowledge EXchange and access (XS) (INFRASTRUCTURESF7-2010-262067/PRIME-XS), and the Lundbeck Foundation (R48-A4649). SAW is supported by a postdoctoral grant from The Danish Council for Independent Research (FSS: 10-085134). The Center for Protein Research is supported by a grant from the Novo Nordisk Foundation.

## Author contributions

BTW and CC designed the project, BTW performed majority of mass spectrometry experiments and data analysis, VI and CS helped with MS analysis and yeast experiments, SAW helped with data analysis, TM and CM analyzed acetyl-CoA concentrations, BTW, RZ, and CC wrote the manuscript, all authors read and commented on the manuscript.

## Conflict of interest

The authors declare they have no conflict of interest.

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
