## [Review Process File · Molecular Systems Biology]

Acetylation Dynamics and Stoichiometry in *Saccharomyces cerevisiae*

Brian T. Weinert, Vytautas Iesmantavicius, Tarek Moustafa, Christian Schölz, Sebastian A. Wagner, Christoph Magnes, Rudolf Zechner, Chunaram Choudhary

Corresponding author: Chunaram Choudhary, University of Copenhagen

Review timeline:	Submission date:	06 August 2013
	Editorial Decision:	05 September 2013
	Revision received:	06 November 2013
	Accepted:	11 December 2013

Editor: Thomas Lemberger

Transaction Report:

1st Editorial Decision

05 September 2013

Thank you again for submitting your work to Molecular Systems Biology. We have now heard back from the three referees who agreed to evaluate your manuscript. As you will see from the reports below, the referees find the topic of your study interesting. They raise, however, several concerns on your work, which should be convincingly addressed in a revision of this work. The recommendations provided by the reviewers are very clear in this regard and refer to the need of further experimental data to strengthen the major conclusions of the study. In particular, the validation of the efficiency of the global chemical acetylation protocol (Reviewer #1 point 1, reviewer #3, point 4) appears to be important to consolidate the conclusion with regard to the stoichiometry of acetylation.

We would also kindly ask you to deposit your proteomics data in an appropriate public repository and indicate the relevant accession number/link/identifier in a sub-section entitled 'Data availability' at the end of the Materials & Method section.

If you feel you can satisfactorily deal with these points and those listed by the referees, you may wish to submit a revised version of your manuscript. Please attach a covering letter giving details of the way in which you have handled each of the points raised by the referees. A revised manuscript will be once again subject to review and you probably understand that we can give you no guarantee at this stage that the eventual outcome will be favorable.

Reviewer #1 :

Weinert and colleagues used a SILAC-based proteomics approach to study the dynamics and stoichiometry of lysine acetylation in yeast in response to the growth conditions and the metabolic enzymes for acetyl-CoA generation. This is extended work from their early studies on yeast acetylome published in *Mol Cell Proteomics*, 2012, 11, 1510-22. They showed that a majority of acetylation sites in yeast, particularly mitochondrial protein, significantly increased in stationary phase as compared to exponential phase, whereas the protein and phosphorylation abundance did not show global change. They next investigated the effects of two key metabolic enzymes for acetyl-CoA generation, *pda1* and *cit1*, and different carbon sources on protein acetylation, showing that protein acetylation is dependent on acetyl-CoA level in distinct cellular compartments. Using bovine serum albumin (BSA), they showed that acetyl-CoA can non-enzymatically acetylate protein at 100 μ M level in vitro. Furthermore, by partial chemical acetylation of yeast proteome using acetyl-phosphate and SILAC-based MS analysis, the authors modified a previously reported method (*Science signaling* 2010, 3, ra3) to calculate the stoichiometries of acetylation sites in yeast. Using this method, they showed that mitochondrial and cytoplasmic acetylation was in low stoichiometry, whereas nuclear acetylation was in high stoichiometry. In general, this is a novel and a piece of very interesting story. However, several issues need be properly addressed prior to its publication:

1. One major issue of this manuscript is the reliability of partial chemical acetylation method for stoichiometry analysis. This calculation is based on the premise that each individual acetylation sites (whatever its real stoichiometry and position) in the yeast proteome will be acetylated by the same (or similar) level (Figure 4A). In this manuscript, the authors assumed that 10% of each Kac site in the yeast proteome will be chemically acetylated by acetyl phosphate. However, due to high order structures of proteins, the complex of whole cell lysate protein, and other factors, evenly chemical acetylation of each individual acetylation site in cellular proteome is almost impossible. The validation experiment by one acetylation peptide, K(ac)VPQVSTPTLVEVSR, from a single purified protein BSA (Fig. S4) is not sufficient to support this conclusion, and the other identified acetylation peptides in BSA did not show the similar change. Additional experimental data are needed to evaluate the chemical acetylation efficiency for the global in vivo acetylation sites by acetyl phosphate. In addition, as the corresponding unmodified peptides were unchanged (Fig. 5C) after partial chemical acetylation, the previous algorithm for absolute stoichiometry calculation reported by Olsen, et al. (*Science signaling* 2010, 3, ra3) cannot be directly applied to such calculation. The author should give a detailed explanation on how the stoichiometries of acetylation sites in Fig. 5D were calculated. It will also be of interest to compare the stoichiometry results of the same acetylation sites calculated from this partial chemical acetylation approach and the previously reported algorithm (*Science signaling* 2010, 3, ra3). In general, though this partial chemical acetylation approach could relatively reflect the trend of acetylation site stoichiometry, the reliability of such calculation for the individual sites remains to be further validated.

2. Another issue is that, in the abstract, the authors concluded that non-enzymatic acetylation may be the major mechanism driving acetylation in cells. There are a few issues with this statement. Although this manuscript showed that the high concentration of acetyl-coA can non-enzymatically acetylate lysine residues, there is no conclusive data in this manuscript to support such conclusion.

3. This manuscript only presented the Gene Ontology functional analysis of high stoichiometry acetylation sites. It is interesting to know the functional analysis of low stoichiometry acetylation sites. Additional comparison and discussion of these findings to the author's previous yeast acetylome findings (*Mol Cell Proteomics*, 2012, 11, 1510-22) will help give further insight of lysine acetylation biology in yeast.

Minor issues:

4. It is not informative that the authors presented the amount (Mole) of acetyl-CoA in yeast cells in Fig. 3A. It should be present in molar concentration rather Mole amount.

5. In page 15, in the method section, why there is no detergent in the lysis buffer? Please clarify. In page 16, please also clarify the recipe of lysis buffer that was used for chemical acetylation of BSA using AcP.

Reviewer #2:

This is an excellent manuscript in which high-end quantitative proteomics is used to address fundamental issues with regard to the role and nature of a major post-translation modification, the acetylation of proteins on lysine residues. The paper is well written, the figures are clear and descriptive, and the extensive supplementary data is well organized and accessible. Choudhary and colleagues provide several lines of evidence indicating that subcellular acetyl-CoA levels determine the extent of protein lysine acetylation. Moreover, their data are consistent with the hypothesis and strengthen the notion that many cellular lysine acetylations occur nonenzymatically. Most importantly, the authors introduce an elegant quantitative MS concept designed to identify low stoichiometry modifications. The key findings here are that most lysine acetylations occur at low or very low stoichiometry, in particular in mitochondria and in the cytosol, whereas high stoichiometry acetylation sites, as they are prominent in the nucleus, are highly enriched for functional relevant sites. Thus, this manuscript may fundamentally change our view on lysine acetylation in the biological context, in a way that the functional role of low stoichiometry (and likely nonenzymatic) modifications is put in question, and that the identification of high stoichiometry sites should be a key element in future acetylomics studies to promote the understanding of functionally relevant modification events. I have no doubt this paper will have a major impact, due to its highly relevant findings and due to its well-conceived experimental concepts. Therefore, I recommend the work by Choudhary and colleagues for publication in *Molecular Systems Biology* once a few minor points have been addressed.

Minor points:

1. Biological replicate data referred to in the manuscript (p. 4) and shown in Fig. S1 is not found in Tables S2 and S3. In Table S4, the table header description for Ratio H/L is incorrect.
2. P. 5 mentioning the loss of Cit1 increased acetylation of mitochondrial protein by an additional 1.8-fold appears to be inconsistent with Fig. 1C (1.2-fold increase).
3. P. 5/6: The reasoning "We found that acetyl-CoA levels were reduced in growth-arrested cells (Figure 3A), suggesting that acetylation occurs due to prolonged exposure to acetyl-CoA." is not clear to me. Wouldn't lower acetyl-CoA level suggest lower non-enzymatic protein acetylation, as observed in cytoplasmic/nuclear compartments (Fig. 2B)?
4. P.7 - Please comment on whether or not chemical modification likely occurs with similar efficiency irrespective of the modification site or protein. Which is the intracellular/subcellular concentration of acetyl-CoA, is it high enough to plausibly explain nonenzymatic lysine acetylation events?

Reviewer #3 :

The authors used quantitative mass spectrometry to analyze acetylation dynamics in *Saccharomyces cerevisiae* by comparing the protein expression, acetylation and phosphorylation level between exponential stage and stationary phase, including wild type and mutant strains. They found that acetylation accumulated in growth-arrested cells in a manner that depended on acetyl-CoA generation in distinct subcellular compartments. Mitochondrial acetylation levels correlated with acetyl-CoA concentration in vivo and acetyl-CoA acetylated lysines nonenzymatically in vitro. They also developed a mass spectrometry based method for the estimation of protein acetylation stoichiometry by chemical acetylation of unmodified peptides with acetyl-phosphate (AcP) and found most acetylation sites are modified with low stoichiometry and the level of acetylation is affected by exposure to acetyl-CoA, suggesting that nonenzymatically acetylation may be the major mechanism driving the acetylation in yeast cells. However, the conclusion was made based on the global pattern of acetylation changes which lacks the analysis of specific proteins and no biological validation was provided. Besides, the accuracy and reliability of the method for acetylation stoichiometry estimation need to be further improved and results need to be validated as there are too many estimations and assumptions.

Major concerns:

1 The authors used SILAC labeling and quantitative mass spectrometry to reveal the lysine acetylation dynamics and $^{12}\text{C}_6$ $^{14}\text{N}_2$ -lysine ("light") or $^{13}\text{C}_6$ $^{15}\text{N}_2$ -lysine (SILAC "heavy") was used to label proteins *in vivo* from different stages for comparison.

It was also mentioned in results section that yeast cells were transferred into lysine lacking media to control the transition from exponentially growing stage to stationary phase. It is doubtful that the lysine deprivation could affect the metabolism of lysine already incorporated into proteins as the protein level was not globally changed. The authors should describe the details of the experiment such as how long the cells were kept in lysine lacking media and check if the increase of acetylation was introduced by the lysine deprivation. The authors may also use other approaches to achieve the stationary phase of yeast cells. Besides, Figure S1 showed that the acetylation sites had much better correlation between two biological replicates than that phosphorylation site end even proteins. It is very rare in the mass spectrometric quantification of PTMs. Larger variation should be observed with modification sites as they were quantified with just one peptide while proteins were quantified with multiple peptides.

2 Acetylation dynamics in growth-arrested yeast cells was studied by statistically comparing the global level of acetylation between different subcellular compartments and nutritional conditions and mutations. Although most of the differences were significant, no examples of specific proteins were provided or validated as the global pattern can be provided with ambiguous conclusion. The authors claimed that the phosphorylation level was not globally increased in growth-arrested yeast cells as the average ratio was 1. However, 50% of the quantified phosphorylation sites showed great changes. When it comes to acetylation, although the ratio of acetylation sites has significantly different pattern as acetylated proteins, the authors didn't compare the ratio of site and the protein containing this sites. It would be better to plot the ratio of site versus the ratio of corresponding protein to show the increasing of acetylation level compared with protein expression change. Besides, some conclusion was based on non-significant differences.

In Figure 2 E, it was claimed that acetate promotes nuclear acetylation. However, the p value was just $8e-7$, which is higher than the criterion ($1e-16$).

3 To prove that acetyl-CoA concentration correlates with the acetylation level in mitochondria, the authors used acetyl-CoA assay to monitor its concentration in yeast cells and the concentration of acetyl-CoA in mitochondria was calculated based on some hypothesis and assumption. It would be straightforward to quantify the acetyl-CoA from isolated mitochondria from same number of cells to compare the concentration in EP and GA stage.

4 For the estimation of acetylation stoichiometry, as the chemical acetylation is not complete and the extent of chemical modification could be peptide sequence dependent, as well as variations between replicates, it is not accurate to use an assumed 10% as the extent of chemical acetylation of each site. The level of chemical acetylation of yeast lysate digest need to be investigated the variance among different peptides also need to be studied. In the majority of quantified site showed L/H ratio higher than 2, it is doubtful that these ratios are accurate as mass spectrometry has limited dynamic range in quantification and larger variations in the ratios between replicates when the log₂ ratio is higher than 2, which affect the accuracy in the estimation of acetylation stoichiometry. As peptide could be completely chemically acetylated by Ac, it would be possible to use synthesized peptides standard and MRM to quantify the absolute stoichiometry of acetylation and validate the results of selected acetylated proteins with important functions. For the sites without SILAC ratios, the SILAC ratios were estimated by calculating the increased intensity of AcP-treated "light" peptides relative to an empirically determined detection limit for naturally occurring "heavy" peptides, the authors need provide the criteria in the peak selecting (intensity, S/N ratio, etc;).

Response to reviewers' comments

We thank the reviewers for carefully reviewing this manuscript and for providing insightful feedback. We were encouraged to read that the reviewers found our work “a novel and a piece of very interesting story” and that “this paper will have a major impact.”

While the reviewers acknowledge the strengths of the manuscript, they have also raised several points. In particular, all three reviewers raised concerns about the degree and variability of the partial chemical acetylation used to estimate acetylation site stoichiometry (reviewer #1, major concern #1, reviewer #2, concern #4, and reviewer #3, major concern #4).

We agree with this criticism, and following their suggestions we have performed additional experiments and revised the manuscript to further clarify how we interpret the partial chemical acetylation and how we estimate stoichiometry. In summary, we have now performed absolute quantification (AQUA) analysis of 8 acetylation sites and compared acetylation stoichiometries observed by this method with our partial chemical acetylation approach. These results show that stoichiometry estimates calculated by these two independent methods are well correlated (Spearman's correlation of -0.92), validating our approach.

This manuscript provides new insights into the dynamics of acetylation and demonstrates that mitochondrial acetylation occurs within this organelle, that acetylation in yeast is regulated in a compartment-specific manner, and provides the first global analysis of acetylation stoichiometry in yeast or any other organism. It is important to consider these fundamental properties of acetylation in investigating the functions of this modification.

Below, first we provide a detailed point-by-point response to reviewers' concerns.

Reviewer #1 (Remarks to the Author):

Weinert and colleagues used a SILAC-based proteomics approach to study the dynamics and stoichiometry of lysine acetylation in yeast in response to the growth conditions and the metabolic enzymes for acetyl-CoA generation. This is extended work from their early studies on yeast acetylome published in *Mol Cell Proteomics*, 2012, 11, 1510-22. They showed that a majority of acetylation sites in yeast, particularly mitochondrial protein, significantly increased in stationary phase as compared to exponential phase, whereas the protein and phosphorylation abundance did not show global change. They next investigated the effects of two key metabolic enzymes for acetyl-CoA generation, *pda1* and *cit1*, and different carbon sources on protein acetylation, showing that protein acetylation is dependent on acetyl-CoA level in distinct cellular compartments. Using bovine serum albumin (BSA), they showed that acetyl-CoA can non-enzymatically acetylate protein at 100 μ M level in vitro. Furthermore, by partial chemical acetylation of yeast proteome using acetyl-phosphate and SILAC-based MS analysis, the authors modified a previously reported method (*Science signaling* 2010, 3, ra3) to calculate the stoichiometries of acetylation sites in yeast. Using this method, they showed that mitochondrial and cytoplasmic acetylation was in low stoichiometry, whereas nuclear acetylation was in high stoichiometry. In general, this is a novel and a piece of very interesting story. However, several issues need be properly addressed prior to its publication:

1. One major issue of this manuscript is the reliability of partial chemical acetylation method for stoichiometry analysis. This calculation is based on the premise that each individual acetylation sites (whatever its real stoichiometry and position) in the yeast proteome will be acetylated by the same (or similar) level (Figure 4A). In this manuscript, the authors assumed that 10% of each Kac site in the yeast proteome will be chemically acetylated by acetyl phosphate. However, due to high order structures of proteins, the complex of whole cell lysate protein, and other factors, evenly chemical acetylation of each individual acetylation site in cellular proteome is almost impossible. The validation experiment by one

acetylation peptide, K(ac)VPQVSTPTLVEVSR, from a single purified protein BSA (Fig. S4) is not sufficient to support this conclusion, and the other identified acetylation peptides in BSA did not show the similar change. Additional experimental data are needed to evaluate the chemical acetylation efficiency for the global in vivo acetylation sites by acetyl phosphate.

The reviewer was correct to point out that our estimate of maximum 10% chemical acetylation was not well supported. Our previous estimate of chemical acetylation of BSA by 100mM AcP was problematic for several reasons; 1) It was not performed in yeast lysates. 2) It was based on label free measurements that are less accurate than quantitative methods such as SILAC. 3) It was based on a separate estimate of acetylation in acetic anhydride-treated BSA and was therefore not directly estimated.

To obtain a more accurate estimate of the degree of chemical acetylation by AcP in the revised version, we have used an AQUA method for calculating stoichiometries of eight different, endogenous acetylation sites, and compared these results with stoichiometries calculated by our partial chemical acetylation approach.

We revised the results section (page 10) as:

“Comparison to heavy-labeled peptide standards indicated that acetylation stoichiometry, after AcP-treatment, was less than 1% at all eight sites, with a median degree of chemical acetylation that was just 0.07% (Table 1, Figure S7A-D). AcP-sensitivity was well-correlated with acetylation stoichiometry as determined by the AQUA method (Spearman’s correlation of -0.92, Figure 5C), confirming our prediction that low stoichiometry sites would be most sensitive to partial chemical acetylation and providing independent validation of our method.

We estimated acetylation stoichiometry based on the conservative assumption that chemical acetylation from 100mM AcP was less than 1% at all sites. A site with 10-fold increased acetylation after AcP-treatment was estimated to have a stoichiometry that is <0.1% while a site with 20-fold increased acetylation was estimated to have a stoichiometry that is <0.05%.”

and (page 11, paragraph 1):

“It was not possible to accurately estimate the stoichiometry of sites that were AcP-insensitive (SILAC ratio L/H <2) as the relative changes in acetylation were of a similar magnitude to the variability of these measurements. Thus, AcP-insensitive sites were estimated to have acetylation stoichiometry that is >1%.”

We agree with the reviewer that the degree of chemical acetylation may vary for different lysines. We acknowledge this limitation in the discussion section.

From the discussion section (page 14, bottom and page 15, top):

“To better understand our estimates of acetylation stoichiometry, it is important to note that individual lysines may vary in the degree of chemical acetylation by AcP. Without knowing the exact degree of chemical acetylation at each position it is not possible to make a precise estimate of acetylation stoichiometry. In order to account for this variability we based our estimates on the conservative assumption that the degree of partial chemical acetylation by AcP was less than 1%. Since we determined that 100mM AcP caused a median chemical acetylation of just 0.07% (between 0.01% and 0.11%), we are likely to overestimate the stoichiometry of acetylation. Thus, stoichiometry estimates are presented as less than (<) values, to designate an estimated stoichiometry that is less than the indicated amount. In addition, some lysines may be inaccessible or otherwise unreactive, resulting in an incorrect estimate of high stoichiometry acetylation. Regardless of these limitations, the results obtained using our approach indicated that it was able to distinguish acetylation stoichiometry at the site level. Absolute quantification indicated that acetylation stoichiometry was inversely proportional to AcP-sensitivity (Table 1, Figure 5C). The highly significant bias to identify high stoichiometry sites on nuclear proteins indicated that the identification of such sites was non-

random. Similarly, sites we predicted to have high stoichiometry were significantly more likely to have known functional roles. Our method further distinguished between low stoichiometry acetylation in the cytoplasm and mitochondria, indicating that the difference in acetylation levels between these two subcellular compartments was greater than the inherent variability of lysine reactivity with AcP. Thus, the resolution of our assay was not limited by the variability in lysine reactivity”

In addition, as the corresponding unmodified peptides were unchanged (Fig. 5C) after partial chemical acetylation, the previous algorithm for absolute stoichiometry calculation reported by Olsen, et al. (Science signaling 2010, 3, ra3) cannot be directly applied to such calculation. The author should give a detailed explanation on how the stoichiometries of acetylation sites in Fig. 5D were calculated. It will also be of interest to compare the stoichiometry results of the same acetylation sites calculated from this partial chemical acetylation approach and the previously reported algorithm (Science signaling 2010, 3, ra3). In general, though this partial chemical acetylation approach could relatively reflect the trend of acetylation site stoichiometry, the reliability of such calculation for the individual sites remains to be further validated.

The reviewer is correct to point out that it is not possible to estimate stoichiometry by the method of Olsen et al if corresponding unmodified peptides were unchanged. For this reason, we were not able to estimate stoichiometry by this method (nor compare results using this method to the method used in our study) and we instead devised the partial chemical acetylation approach used in this manuscript. In this method we measure the relative abundance of acetylated peptides before and after chemical acetylation and estimated stoichiometry based on the relative increase in acetylation and assuming a degree of chemical acetylation that was low (<1%). As pointed out above, we have further validated our approach in the revised manuscript by quantifying the degree of partial chemical acetylation using an independent method (AQUA).

The detailed explanation of how stoichiometries were calculated is further explained above, in response to issue #1.

2. Another issue is that, in the abstract, the authors concluded that non-enzymatic acetylation may be the major mechanism driving acetylation in cells. There are a few issues with this statement. Although this manuscript showed that the high concentration of acetyl-coA can non-enzymatically acetylate lysine residues, there is no conclusive data in this manuscript to support such conclusion

Our data imply that non-enzymatic acetylation may occur in cells, an idea also supported by other recent papers (Moellering and Cravatt, Science. 2013 Aug 2;341(6145):549-53; Wagner and Payne, J Biol Chem. 2013 Oct 4;288(40):29036-45). However, we acknowledge that conclusive evidence for this idea may be impossible to produce (as it would require elimination of possible, yet unknown enzymatic activity). Therefore, we have removed this claim from the abstract.

3. This manuscript only presented the Gene Ontology functional analysis of high stoichiometry acetylation sites. It is interesting to know the functional analysis of low stoichiometry acetylation sites. Additional comparison and discussion of these findings to the author's previous yeast acetylome findings (Mol Cell Proteomics, 2012, 11, 1510-22) will help give further insight of lysine acetylation biology in yeast.

We feel that the Gene Ontology analysis of low stoichiometry sites would not be informative. We showed that the detection of acetylation is highly biased to occur on abundant proteins (Figure 6E), therefore, comparison of low-stoichiometry acetylated proteins to non-acetylated proteins will simply indicate the gene ontology terms associated with abundant proteins. The proper comparison is between low-stoichiometry acetylation and high-stoichiometry acetylation, however the vast majority of acetylation is low stoichiometry (~95%) and it is not possible to see GO term enrichment within a group that constitutes the vast majority of the population. Furthermore, many low-stoichiometry sites occur together with high stoichiometry sites on the same proteins, making it difficult to distinguish proteins with low-stoichiometry

acetylation. In addition, our data suggests that all accessible lysines may be acetylated at a low stoichiometry, making the functional characterization of such acetylation uninformative.

Minor issues:

4. It is not informative that the authors presented the amount (Mole) of acetyl-CoA in yeast cells in Fig. 3A. It should be present in molar concentration rather Mole amount.

The molar concentration of acetyl-CoA in cells is an estimate (which includes several assumptions about cell volume, cell number, and acetyl-CoA recovery), while the values shown in Figure 3A indicate the amounts determined using our assay, we prefer to show the actual measured values rather than estimates based on assumptions.

However, we agree that understanding the molar concentration in cells would be informative in our study and we have therefore added the following paragraph to the manuscript (page 7). “The physiological concentration of acetyl-CoA in yeast is not well-studied. One study estimated acetyl-CoA levels (3-30 μ M) in nutrient-starved yeast undergoing metabolic cycles {Cai, 2011 #151}, conditions that contrast with the excess of glucose and high growth rates of yeast grown on synthetic complete (SC) media in our study. We estimated a similar cellular concentration of \sim 30 μ M acetyl-CoA in exponentially growing cells. However, this estimate assumes complete recovery of acetyl-CoA, and is therefore likely to underestimate the actual cellular concentration. We showed that mitochondrial acetyl-CoA was \sim 20 to 30-fold higher than non-mitochondrial acetyl-CoA, suggesting a concentration of acetyl-CoA in mitochondria that approaches the millimolar range (\sim 0.5 to 1mM based on our estimates). This estimate is consistent with previous work showing that acetyl-CoA can reach millimolar levels in rat liver mitochondria {Garland, 1965 #258}.”

5. In page 15, in the method section, why there is no detergent in the lysis buffer? Please clarify. In page 16, please also clarify the recipe of lysis buffer that was used for chemical acetylation of BSA using AcP.

Lysis buffer is prepared without detergent and detergent is added after cell pulverization. This is to ensure efficient breakage of the frozen cell suspension. This is clarified in the manuscript “Cells were harvested at the indicated time points, washed once with sterile H₂O, and resuspended in lysis buffer (50mM Tris, pH7.5, 150mM NaCl, 1mM EDTA, 1x mini complete protease inhibitor cocktail (Roche), 5mM sodium fluoride, 1mM sodium orthovanadate, 5mM beta-glycerophosphate, 10mM nicotinamide, and 5 μ M tricostatin A) at \sim 50 OD₆₀₀ cells/ml lysis buffer. The cell suspension was frozen drop-wise in liquid nitrogen and ground in a liquid nitrogen chilled steel container by the Retsch MM 400 Ball Mill for 5min. at 25Hz. The lysate was thawed, NP-40 and sodium deoxycholate were added to a final concentration of 1% and 0.1%, respectively, and clarified by centrifugation.”

The following was added to clarify the buffer composition used to treat BSA with AcP (page 18). “For treatment with AcP, BSA was prepared in yeast lysis buffer (as above, with NP-40 and deoxycholate added) at a concentration of 20mg/ml.”

Reviewer #2 (Remarks to the Author):

This is an excellent manuscript in which high-end quantitative proteomics is used to address fundamental issues with regard to the role and nature of a major post-translation modification, the acetylation of proteins on lysine residues. The paper is well written, the figures are clear and descriptive, and the extensive supplementary data is well organized and accessible. Choudhary and colleagues provide several lines of evidence indicating that subcellular acetyl-CoA levels determine the extent of protein lysine acetylation. Moreover, their data are consistent with the hypothesis and strengthen the notion that many cellular lysine acetylations occur nonenzymatically. Most importantly, the authors introduce an elegant quantitative MS

concept designed to identify low stoichiometry modifications. The key findings here are that most lysine acetylations occur at low or very low stoichiometry, in particular in mitochondria and in the cytosol, whereas high stoichiometry acetylation sites, as they are prominent in the nucleus, are highly enriched for functional relevant sites. Thus, this manuscript may fundamentally change our view on lysine acetylation in the biological context, in a way that the functional role of low stoichiometry (and likely nonenzymatic) modifications is put in question, and that the identification of high stoichiometry sites should be a key element in future acetylomics studies to promote the understanding of functionally relevant modification events. I have no doubt this paper will have a major impact, due to its highly relevant findings and due to its well-conceived experimental concepts. Therefore, I recommend the work by Choudhary and colleagues for publication in *Molecular Systems Biology* once a few minor points have been addressed.

Minor points:

1. Biological replicate data referred to in the manuscript (p. 4) and shown in Fig. S1 is not found in Tables S2 and S3. In Table S4, the table header description for Ratio H/L is incorrect.

We thank the reviewer for pointing this out, and apologize for this oversight. It is fixed in the revised manuscript.

2. P. 5 mentioning the loss of Cit1 increased acetylation of mitochondrial protein by an additional 1.8-fold appears to be inconsistent with Fig. 1C (1.2-fold increase).

This comment appears to result from a misunderstanding. The increase in acetylation in exponential phase cit1 cells in Figure 2C is 1.2, as the reviewer indicates. However, the increase in growth-arrested cit1 cells (Figure 2D) is 1.8-fold, and is correctly indicated in the manuscript. "Loss of Pda1 completely suppressed the increased acetylation of mitochondrial proteins in growth-arrested cells while loss of Cit1 further increased the acetylation of mitochondrial proteins (an additional 1.8-fold) in growth-arrested cells (Figure 2D, Figure S2C, and Table S7)."

3. P. 5/6: The reasoning "We found that acetyl-CoA levels were reduced in growth-arrested cells (Figure 3A), suggesting that acetylation occurs due to prolonged exposure to acetyl-CoA." is not clear to me. Wouldn't lower acetyl-CoA level suggest lower non-enzymatic protein acetylation, as observed in cytoplasmic/nuclear compartments (Fig. 2B)?

We agree with the reviewer that these data appear inconsistent; however it is extremely difficult to verify that the recovery of acetyl-CoA from exponentially growing and growth-arrested cells is equally efficient. Therefore, it is difficult to make a conclusive statement based on these data, we have further expanded on this point in the revised manuscript "However, this difference may be due to reduced recovery of acetyl-CoA from growth-arrested cells, which have a substantially increased cell wall and are known to be refractory to cell lysis. We consistently recovered less protein from growth-arrested cells (data not shown), suggesting that these cells were more difficult to lyse, or that cell size and/or protein content was reduced under these conditions. Such differences in cell physiology may explain the lower amount of acetyl-CoA per OD₆₀₀ of growth-arrested cells as determined by our assay." Regardless, acetylation did increase in growth-arrested cells, suggesting that prolonged exposure to acetyl-CoA under these conditions may be the mechanism driving this increase.

4. P.7 - Please comment on whether or not chemical modification likely occurs with similar efficiency irrespective of the modification site or protein. Which is the intracellular/subcellular concentration of acetyl-CoA, is it high enough to plausibly explain nonenzymatic lysine acetylation events?

With regards to the efficiency of chemical acetylation, please see our response reviewer #1, point #1.

With regards to acetyl-CoA concentration, a discussion of subcellular concentration has been added to the manuscript, please refer to our response to reviewer #1, minor point #4.

Reviewer #3 (Remarks to the Author):

The authors used quantitative mass spectrometry to analyze acetylation dynamics in *Saccharomyces cerevisiae* by comparing the protein expression, acetylation and phosphorylation level between exponential stage and stationary phase, including wild type and mutant strains. They found that acetylation accumulated in growth-arrested cells in a manner that depended on acetyl-CoA generation in distinct subcellular compartments. Mitochondrial acetylation levels correlated with acetyl-CoA concentration in vivo and acetyl-CoA acetylated lysines nonenzymatically in vitro. They also developed a mass spectrometry based method for the estimation of protein acetylation stoichiometry by chemical acetylation of unmodified peptides with acetyl-phosphate (AcP) and found most acetylation sites are modified with low stoichiometry and the level of acetylation is affected by exposure to acetyl-CoA, suggesting that nonenzymatically acetylation may be the major mechanism driving the acetylation in yeast cells. However, the conclusion was made based on the global pattern of acetylation changes which lacks the analysis of specific proteins and no biological validation was provided. Besides, the accuracy and reliability of the method for acetylation stoichiometry estimation need to be further improved and results need to be validated as there are too many estimations and assumptions.

Major concerns:

1 The authors used SILAC labeling and quantitative mass spectrometry to reveal the lysine acetylation dynamics and $^{12}C_6$ $^{14}N_2$ -lysine ("light") or $^{13}C_6$ $^{15}N_2$ -lysine (SILAC "heavy") was used to label proteins in vivo from different stages for comparison. It was also mentioned in results section that yeast cells were transferred into lysine lacking media to control the transition from exponentially growing stage to stationary phase. It is doubtful that the lysine deprivation could affect the metabolism of lysine already incorporated into proteins as the protein level was not globally changed. The authors should describe the details of the experiment such as how long the cells were kept in lysine lacking media and check if the increase of acetylation was introduced by the lysine deprivation. The authors may also use other approaches to achieve the stationary phase of yeast cells.

The manuscript has been edited to indicate that cells were growth-arrested for ~24 hours (page 5, middle). Cells were arrested with or without lysine present and acetylation was similarly increased under both conditions, indicating that lysine deprivation was not specifically responsible for the increase in acetylation (data not shown). Furthermore, we showed that lysine depletion alone was not sufficient to cause increased acetylation as glucose (or acetate) was required for this effect (Figure 2).

Besides, Figure S1 showed that the acetylation sites had much better correlation between two biological replicates than that phosphorylation site end even proteins. It is very rare in the mass spectrometric quantification of PTMs. Larger variation should be observed with modification sites as they were quantified with just one peptide while proteins were quantified with multiple peptides.

The better correlation is likely due to the robust and nearly comprehensive (most sites change in abundance) changes seen in acetylation, since most proteins did not change in abundance the correlation is more affected by the inherent variability of these measurements. In other words the magnitude of acetylation changes is much greater than the inherent variability of these measurements, thus the correlation is higher. While these correlations differ, they are all significant.

2 Acetylation dynamics in growth-arrested yeast cells was studied by statistically comparing the global level of acetylation between different subcellular compartments and nutritional conditions and mutations.

Although most of the differences were significant, no examples of specific proteins were provided or validated as the global pattern can be provided with an ambiguous conclusion.

Validation of individual proteins is complicated by the extremely low-level of acetylation. In the revised manuscript we validate stoichiometries of eight individual sites from two different proteins using the AQUA approach (Figure 5C, Figure S7). These results further confirmed our acetylation stoichiometry estimates.

The authors claimed that the phosphorylation level was not globally increased in growth-arrested yeast cells as the average ratio was 1. However, 50% of the quantified phosphorylation sites showed great changes.

We noted that phosphorylation was highly regulated in response to stationary phase and that these changes occurred on proteins associated with Gene Ontology terms that indicated specific biological processes were regulated. We indicate this in the manuscript as: "Furthermore, Gene Ontology (GO) enrichment analysis of protein and phosphorylation site changes indicated both up-regulation and down-regulation of specific processes in stationary phase cells (Figures S1D and E)."

It is clear that phosphorylation should be, and is, regulated under these conditions. Many of the affected phosphorylation sites occur on proteins that are known to be regulated under conditions of nutrient starvation; however, the comparison of phosphorylation changes was only included to show that phosphorylation was regulated (both in terms of increase and decrease in the modification) in a site-specific manner and not globally increased in the same manner as acetylation. We have added additional text to the manuscript to clarify that changes in protein and phosphorylation abundance are likely to result from a regulated physiological response (new text underlined). "In contrast, protein and phosphorylation abundance, while affected in stationary phase cells, was not globally increased (Figure 1), indicating that stationary phase did not cause the accumulation of proteins or PTMs generally. Furthermore, Gene Ontology (GO) enrichment analysis of protein and phosphorylation site changes revealed both up-regulation and down-regulation of specific processes in stationary phase cells (Figures S1D and E), suggesting that such changes occurred in a regulated manner."

When it comes to acetylation, although the ratio of acetylation sites has significantly different pattern as acetylated proteins, the authors didn't compare the ratio of site and the protein containing this sites. It would be better to plot the ratio of site versus the ratio of corresponding protein to show the increasing of acetylation level compared with protein expression change.

The acetylation site changes shown in Figure 1 were corrected for differences in protein abundance; this was not indicated in the manuscript but has been indicated in the figure legend of the revised manuscript (page 32). Therefore, the ratio changes that are shown take into account differences in protein abundance.

Besides, some conclusion was based on non-significant differences. In Figure 2 E, it was claimed that acetate promotes nuclear acetylation. However, the p value was just $8e-7$, which is higher than the criterion ($1e-16$).

While an asterisk is often used to indicate a significant change (typically p value <0.01), in our case we used an asterisk to indicate significance that was extremely high (p values less than $2e-16$). A p value of $8e-7$ is very significant regardless. Asterisks have been removed from all figures and the p values shown directly (Figures 1, 2, and S2).

3 To prove that acetyl-CoA concentration correlates with the acetylation level in mitochondria, the authors used acetyl-CoA assay to monitor its concentration in yeast cells and the concentration of acetyl-CoA in mitochondria was calculated based on some hypothesis and assumption. It would be straightforward to quantify the acetyl-CoA from isolated mitochondria from same number of cells to compare the concentration in EP and GA stage.

Measurement of compartment-specific pools of acetyl-CoA is an outstanding question in the acetylation field, however, due to technical limitations it is not straightforward (Wellen and Thompson, Nat Rev Mol Cell Biol. 2012 Mar 7;13(4):270-6) . In order to measure acetyl-CoA from isolated mitochondria one must digest the yeast cell wall using zymolase enzyme. This procedure is not effective in growth-arrested cells as the cell wall becomes substantially thicker. Furthermore, the zymolase reaction is typically carried out at 30 degrees for at least 30 minutes (longer for growth-arrested cells), which would likely impact cellular physiology and acetyl-CoA levels. Furthermore, the comparison of different growth states is complicated by physiological changes in cell wall composition, and a direct comparison of mitochondria and whole cells is complicated by both the efficiency of mitochondrial recovery and the differing efficiencies of acetyl-CoA recovered from whole cells versus purified mitochondria.

The following text was added to further clarify the difficulty in comparing exponential phase and stationary phase (growth-arrested) cells. “We consistently recovered less protein from growth-arrested cells (data not shown), suggesting that these cells were more difficult to lyse, or that cell size and/or protein content was reduced under these conditions. Such differences in cell physiology may explain the lower amount of acetyl-CoA per OD600 of growth-arrested cells as determined by our assay.”

One of our key findings was that acetylation stoichiometry and acetyl-CoA concentration was higher in mitochondria, irrespective of growth phase. The comparison of EP and GA cells is not necessary to support this finding. Under the growth conditions used in our study, mutation of Pda1 should abolish mitochondrial acetyl-CoA. Thus, our estimate of mitochondrial acetyl-CoA levels based on comparing Pda1 and wild type cells should be directly comparable. Furthermore, we state that mitochondrial acetyl-CoA constitutes at least 1/3 of the total acetyl-CoA pool, since loss of Pda1 may not have removed all acetyl-CoA from mitochondria. Taking this into consideration, these data indicate that mitochondrial acetyl-CoA levels could be higher than our estimate, but are unlikely to be lower.

4 For the estimation of acetylation stoichiometry, as the chemical acetylation is not complete and the extent of chemical modification could be peptide sequence dependent, as well as variations between replicates, it is not accurate to use an assumed 10% as the extent of chemical acetylation of each site. The level of chemical acetylation of yeast lysate digest need to be investigated the variance among different peptides also need to be studied.

We agree with this criticism and had previously acknowledged this in the discussion section.

While we found that chemical acetylation was variable (median 0.07%, range 0.01% to 0.11%), we also showed that chemical acetylation was highly correlated with AcP-sensitivity (Spearman’s correlation of -0.92) with a statistical significance of 0.001 (which is good considering it is based on only 8 data points). While the degree of chemical acetylation may vary, the greatest degree of chemical acetylation was ~0.1%, thus our estimates of acetylation stoichiometry based on less than 1% chemical acetylation should be accurate even if the variability was much greater than determined by AQUA.

In addition, the variability of chemical acetylation must be less than the differences in acetylation stoichiometry between different classes of proteins, as we were able to resolve these differences using our approach. This is now more clearly noted in the discussion section (page 15) “Regardless of these limitations, the results obtained using our approach indicated that it was able to distinguish acetylation stoichiometry at the site level. Absolute quantification indicated that acetylation stoichiometry was inversely proportional to AcP-sensitivity (Table 1, Figure 5C). The highly significant bias to identify high stoichiometry sites on nuclear proteins indicated that the identification of such sites was non-random. Similarly, sites we predicted to have high stoichiometry were significantly more likely to have known functional roles. Our method further distinguished between low stoichiometry acetylation in the cytoplasm and mitochondria, indicating that the difference in acetylation levels between these two subcellular compartments was greater than the inherent

variability of lysine reactivity with AcP. Thus, the resolution of our assay was not limited by the variability in lysine reactivity.”

We agree that it is important to communicate the limitations of the method used to estimate stoichiometry in our study. However, the results of our analysis indicate that this method is able to distinguish differences in acetylation stoichiometry, an idea that is further supported by the good correlation between stoichiometry determined by AQUA and AcP-sensitivity (Figure 5C), the correlation between abundance-corrected acetylated peptide intensities and AcP sensitivity (Figure 5F), and our finding that distinct classes of proteins were differentially affected by AcP-treatment (Figure 6).

In the majority of quantified site showed L/H ratio higher 24, it is doubtful that these ratios are accurate as mass spectrometry has limited dynamic range in quantification and larger variations in the ratios between replicates when the log₂ ratio is higher than 2, which affect the accuracy in the estimation of acetylation stoichiometry. As peptide could be completely chemically acetylated by Ac, it would be possible to use synthesized peptides standard and MRM to quantify the absolute stoichiometry of acetylation and validate the results of selected acetylated proteins with important functions.

As mentioned above, we have now determined absolute stoichiometry for several acetylation sites. It may be that large relative changes are underestimated by SILAC, however, we are unaware of any studies that demonstrate this to be the case. In addition, our observed ratio L/H changes were highly correlated with our stoichiometry estimates based on AQUA (Figure 5C), suggesting that the measured ratios accurately reflect the acetylation stoichiometry.

For the sites without SILAC ratios, the SILAC ratios were estimated by calculating the increased intensity of AcP-treated "light" peptides relative to an empirically determined detection limit for naturally occurring "heavy" peptides, the authors need provide the criteria in the peak selecting (intensity, S/N ratio, etc;).

The method for determining the detection limit for “heavy” labeled peptides was detailed in the methods section. “The detection limit for naturally occurring acetylated peptides used to calculate minimum increased acetylation in Figure 5G was determined by ranking the observed "heavy" labeled peptides by peptide intensity and calculating the median intensity of the bottom 10%. This intensity value was an order of magnitude higher than the least intense "heavy" peptide observed in our experiments, thus the estimates of increased acetylation based on this detection limit are conservative.”

Furthermore, our results using these ratios was independently confirmed by the distribution of I/iBAQ-A values shown in Figure 5F, which showed that sites without ratios had the lowest median distribution of I/iBAQ-A, further suggesting a lower median stoichiometry at these sites compared to sites with ratios >2.

Acceptance letter

11 December 2013

Thank you again for sending us your revised manuscript. We are now satisfied with the modifications made and I am pleased to inform you that your Report has been accepted for publication.

Thank you very much for submitting your work to Molecular Systems Biology.

Reviewer #3:

The authors' reply is satisfactory and they have addressed the major of the stoichiometry calculation by absolute quantification. The manuscript can be published as it is.